# Unraveling the causal genes and transcriptomic determinants of human telomere length

Ying Chang[1,13], Yao Zhou[2,13], Junrui Zhou[3,4,13], Wen Li[1], Jiasong Cao[1], Yaqing Jing[3], Shan Zhang[3], Yongmei Shen[1], Qimei Lin[1], Xutong Fan[2], Hongxi Yang[2,5], Xiaobao Dong [3], Shijie Zhang[5], Xianfu Yi [2], Ling Shuai [6], Lei Shi [7], Zhe Liu [7], Jie Yang [7], Xin Ma [8], Jihui Hao[9], Kexin Chen[10], Mulin Jun Li [2,10] ✉, Feng Wang [3,11,12] ✉ & Dandan Huang[5,8] ✉

Telomere length (TL) shortening is a pivotal indicator of biological aging and is associated with many human diseases. The genetic determinates of human TL have been widely investigated, however, most existing studies were conducted based on adult tissues which are heavily influenced by lifetime exposure. Based on the analyses of terminal restriction fragment (TRF) length of telomere, individual genotypes, and gene expressions on 166 healthy placental tissues, we systematically interrogate TL-modulated genes and their potential functions. We discover that the TL in the placenta is comparatively longer than in other adult tissues, but exhibiting an intra-tissue homogeneity. Trans-ancestral TL genome-wide association studies (GWASs) on 644,553 individuals identify 20 newly discovered genetic associations and provide increased polygenic determination of human TL. Next, we integrate the powerful TL GWAS with placental expression quantitative trait locus (eQTL) mapping to prioritize 23 likely causal genes, among which 4 are functionally validated, including *MMUT*, *RRM1*, *KIAA1429*, and *YWHAZ*. Finally, modeling transcriptomic signatures and TRF-based TL improve the prediction performance of human TL. This study deepens our understanding of causal genes and transcriptomic determinants of human TL, promoting the mechanistic research on fine-grained TL regulation.

Telomeres are DNA and protein complexes that protect the ends of chromosomes, yet degradative processes that shorten telomeric DNA can lead to loss of telomere function and genomic instability[1,2]. Telomeres shorten with each round of DNA replication in the organism's aging process[3]. Thus, telomere length (TL) has been recognized as a critical indicator of cellular senescence, biological aging, and disease progression[4–6]. In previous studies, determinants of human TL have been extensively interrogated, including different genetic, environmental, and lifestyle factors[2,7]. For example, genetic variants associated with TL have been systematically identified through family studies[8,9]

and genome-wide association studies (GWASs)[10–12]. Nongenetic factors, such as cigarette smoking[13], alcohol consumption[14], and endurance training[15], could modulate telomere attrition processes. In addition, a large-scale cross-tissue TL analysis revealed that TL varied across tissue types and was the shortest in whole blood[16]. Despite these successes, most of the current studies on human TL were performed on postnatal or adult tissues, which confounds the understanding of independent determinates from genetic or environmental factors, especially for non-Medawarian tissues affected by unobserved lifetime exposures[17,18].

Current TL GWASs have uncovered more than a hundred genomic loci significantly associated with leukocyte TL[10–12], however, the true causal genes underlying these polygenic determinants of TL remain elusive. The majority of the identified associations lie in the non-coding regions of the human genome, suggesting that causal variants could influence TL via gene regulatory codes. Expression quantitative trait loci (eQTLs) analysis using gene expression as a key intermediate molecular phenotype, improves the functional interpretation of GWAS findings[19]. Thus, integrating powerful TL GWAS results with eQTLs on tissue rarely affected by non-genetic factors would enhance the TL-causal gene discovery[20,21] and also facilitate accurate TL prediction by incorporating both genomic and transcriptomic information.

In the present study, by assuming that telomere is less affected by extraplacental exposure or other non-genetic effects in the placenta, we profiled terminal restriction fragment (TRF) length of telomere, genotypes, and gene expression on 166 healthy placental tissues. We found that placental TL expresses the intratissue homogeneity and is relatively longer among different human tissues, except for testis. The analysis of gene expression association with placental TL revealed several unique telomere-maintaining patterns. Importantly, we integrated three large-scale TL GWASs from worldwide cohorts and performed a trans-ancestral meta-analysis. Then, placental eQTL mapping and complementary statistical approaches were leveraged to comprehensively prioritize the putative causal genes affecting TL phenotype. We also experimentally validated several newly discovered TL-causal genes in vitro. Finally, we developed an accurate TL prediction model that outperforms the existing strategies.

## Results

### Placental TL demonstrates intra-tissue homogeneity and is relatively longer than in other adult tissues

To measure the telomere-associated phenotypes of human tissues at the early life stage, we conducted population-scale southern blots of TRFs[22], an accurate characterization of telomeres, on 166 healthy placental tissues (see methods for details). These placentae were collected within 10 min of a vaginal delivery from full-term singleton pregnancies ($37^{+0}$–$41^{+6}$ weeks) (Fig. 1A), and none of the participants had any medical disorders or adverse pregnancy outcomes. The average age of the participants was $32 \pm 4.0$ years, and no tobacco-smoking or alcohol-drinking behavior was noted. Among these newborns, 73 were females and 93 were males. No significant differences were observed in maternal age, maternal body mass index (BMI), infant weight, and gestation weeks with regard to infant gender (all $P$ values > 0.05) (Supplementary Data 1). In addition to these placental tissues, genotypes [the Infinium Asian Screening Array (ASA), $n = 166$] and gene expression [RNA sequencing (RNA-seq), $n = 166$] were also profiled for in-depth analysis of the causal genes and transcriptional determinants of placental TL (Fig. 1A).

Previous studies reported that the TRF length of telomere is highly synchronized among various tissues at birth, whereas in adults, TL across tissue types varies within individuals[16,23,24]. The sampling of eight symmetrical sets of placental tissue on the fetal side and maternal side revealed a consistent distribution of telomere fragments, suggesting a homogeneous telomere status across the whole placenta (Fig. 1B). However, compared to intraplacental telomere status, the ranges and variances in TRF lengths were enlarged as assessed by randomly sampling of 15 unrelated individuals (Supplementary Fig. 1). These results implied that the telomere content of placental tissue varies among individuals and could be attributed to varied genetic conditions and intrauterine environments. To evaluate the homogeneity of our measurements and to identify potential sampling errors, we conducted repeated measurements on different regions of the same placental sample three additional times (Supplementary Fig. 2A). This approach allowed us to assess the uniformity of telomere status across the entire placenta and provided critical insights into

possible measurement and sampling errors. Our findings indicated that the quantitative analysis error did not surpass 10% between different experimental batches (Fig. 1C), underscoring the reliability of our TRF measurements. To further ensure the robustness of our data analysis, we employed the TeloTool[25] software with a stringent fit threshold setting. This rigorous criterion enabled us to include only those length determinations that met the predefined standards in our final results, thereby minimizing any potential bias or errors during data analysis (Supplementary Fig. 2B).

Since the southern blot of TRF contains abundant information on telomere content, based on TRF analysis of the fetal side placental tissues from 166 unrelated singleton pregnancies, we quantified the average TRF length (aTL), relative TL (RTL), and short telomere proportion (STP), respectively (see methods for details). Briefly, the RTL was used to evaluate the aTL relative to a standard reference, and the STP measured the percentage of the telomere shorter than 5 kb over fragments, indicating severe telomere damage or wear[26]. Placental aTL ranges from 7.95–17.85 kb among 166 placental tissues (mean = 11.83 kb). Leveraging aTL data of five different adult tissues revealed that aTL in the placenta is longer than that of rest of other tissues, including blood ($N = 231$, mean = 9.44 kb), skin ($N = 12$, mean = 8.73 kb), heart ($N = 6$, mean = 8.69 kb), and lung ($N = 6$, mean = 9.15 kb) (Fig. 1D). This conforms to the expectation that TL is associated with cellular senescence; it is maximal at birth and decreases with age and exposure[27]. No clear association was observed between RTL and the collected demographic factors, including maternal age, gestational days, maternal BMI, infant weight, and placental size (Supplementary Fig. 3). Notably, neonatal sex showed weak evidence of association with RTL in the placenta, wherein males have longer RTL than females ($P$-value = 0.032, $t$-test, Supplementary Fig. 3). Given the limited sample size of our existing TRF measurements in newborns[23,28], the underpowered associations require further ascertainment on large-scale samples. Moreover, the current data revealed that placental STP was negatively correlated with RTL (Fig. 1E), indicating the dependency between TL and short telomere, and RTL could partially explain telomere damage. However, we did not observe significant differences between placental STP and any collected demographic factors (Supplementary Fig. 4). Taken together, the intra-tissue homogeneity of telomere content, long RTL across tissues, as well as less external environmental intervention make placental tissue an ideal proxy for studying the genetic determinants and causal genes of human TL.

### Trans-ancestral GWAS reveals increased polygenic determination of TL

In order to explore the extent of genetic contribution on TL, we first integrated three large-scale TL GWASs from different cohorts on worldwide populations, including Singapore Chinese Health Study (SCHS)[10], NHLBI Trans Omics for Precision Medicine (TOPMed)[12], and UK Biobank (UKBB)[11]. Next, we conducted a trans-ethnic meta-analysis based on these leukocyte TL GWASs, containing 644,553 participants from five human subpopulations (including European, African, East Asian, South Asian, and Hispanic/Latino) (Fig. 2A, B and Supplementary Data 2) (see methods for details). A total of 220 sentinel common variants ($R^2 < 0.01$ between sentinels, minor allele frequency (MAF) ≥ 0.01 across separate GWAS cohorts) were associated with leukocyte TL at genome-wide statistical significance threshold ($P$ value < 5E-8), of which 20 variants were new ($R^2 < 0.01$ with previously documented sentinels), and the remaining variants were originally reported or associated with the sentinels in the three GWASs (Supplementary Data 3 and Supplementary Data 4). We successfully validated 86% of the initially reported loci from the three GWAS resources (Supplementary Data 5). However, those signals not reported in our trans-ethnic GWAS may be attributable to our concentrated focus on common variations. Additionally, the heterogeneity of variant effects and discrepancies in TL measurements might also have played significant

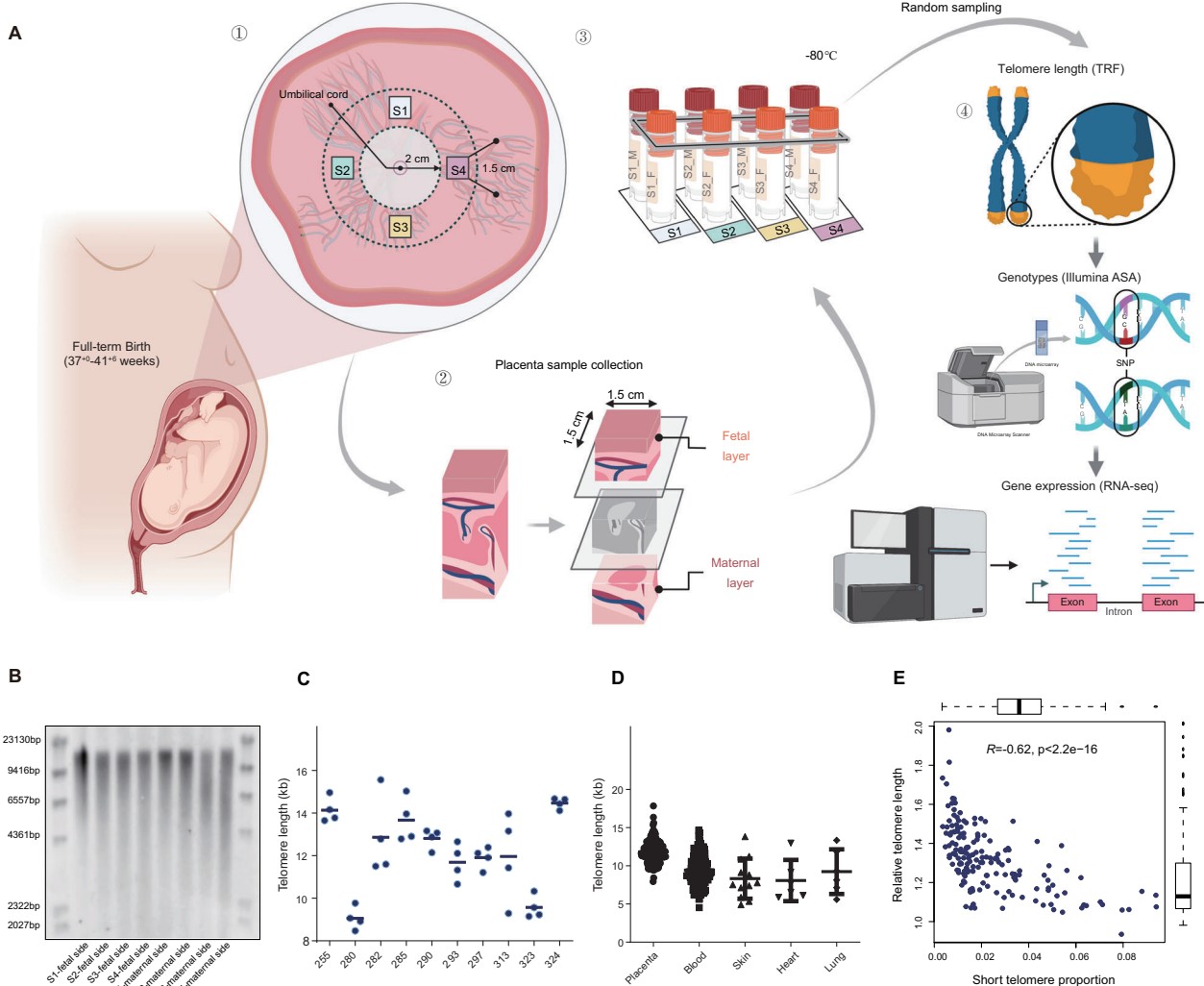

**Fig. 1 | Placental material extraction and measurement. A** Schematic depicts the collection procedures of placental samples and the workflow of analysis in this study. TRF analysis, genotyping, and RNA-seq were conducted simultaneously for each sample. Created with BioRender.com [https://www.biorender.com/]. **B** TL of different sets from the same placental tissue was analyzed by TRF with regular electrophoresis gels ($n = 4$). **C** The statistical analysis presents the telomere length measurements of the same placental sample obtained from different positions and different batches of measurements ($n = 4$). **D** Scatter dot plots of individual data points and mean and standard deviation (SD) showing the distribution of RTL across placenta and 4 different adult tissues. Data are presented as mean values +/- SD ($n = 166$ for placenta, $n = 231$ for blood, $n = 12$ for skin, $n = 6$ for heart, $n = 6$ for lung). **E** Scatter plot shows the correlations between short telomere proportion and RTL ($n = 166$), with a simple linear regression line fitted. For boxplots, five-number summary of the data set (minimum, lower quartile, median, upper quartile and maximum) and outliers are shown. The two-sided *P*-value of Pearson's correlation test was 1.48e-18.

roles. Among the newly discovered loci, the most significant sentinel variant rs10798002 received moderate signals in UKBB and TOPMed cohorts and reached genome-wide significance after meta-analysis, but there was no evidence of the effect of heterogeneity across cohorts ($P_{het} = 0.899$, $I^2 = 0$) (Fig. 2C). This variant is located in the *SWT1* gene, affecting the surveillance of nuclear messenger ribonucleoprotein particles[29]. Gene-set enrichment analysis of these TL GWAS signals identified that the most significantly associated pathways were related to telomere maintenance, telomere organization, and telomere maintenance via telomere lengthening (Fig. 2D), which is consistent with the previous findings[11] and implies that TL GWAS signals were related to the regulation of telomere maintenance.

The trans-ancestral TL GWAS meta-analysis, with the largest sample size to date, provided an effective resource to test the agreement between the genetic determination of TL and observed TL, especially that from fetal tissues under minimal extrauterine intervention. Thus, we genotyped and imputed 6,091,762 genetic variants for the 166 placental samples and performed polygenic risk score

(PRS) analyses based on the TL meta-analysis results. The estimated PRS score of TL was significantly correlated with placental TL measured in our study ($r = 0.21$, *P* value = 0.007) (Fig. 2E), suppressing the correlations using GWAS summary statistics of the single cohort (Supplementary Fig. 5). This suggested that the trans-ancestral GWAS boosts the predictive power of PRS on TL. Also, no significant differences were observed in the distribution of the individual PRS with respect to sex or maternal age (Supplementary Fig. 6). SNP-level associations for each of the significant TL GWAS hits with placental RTL was provided in Supplementary Data 6. In contrast, the evaluation of such correlations based on 442 GTEx whole blood samples revealed a weak association between TL PRS and tested TL measured by biochemical assays (Fig. 2E). Since GTEx applied a Luminex-based method to estimate RTL, the suboptimal measurement may undermine the consistency between PRS-predicted and assayed TL compared to southern blot analysis of TRFs. Additionally, we estimated PRS score of TL based on GWAS hits from the trans-ethnic analysis that showed nominal (*P* value < 0.05) association in the SCHS study. The result

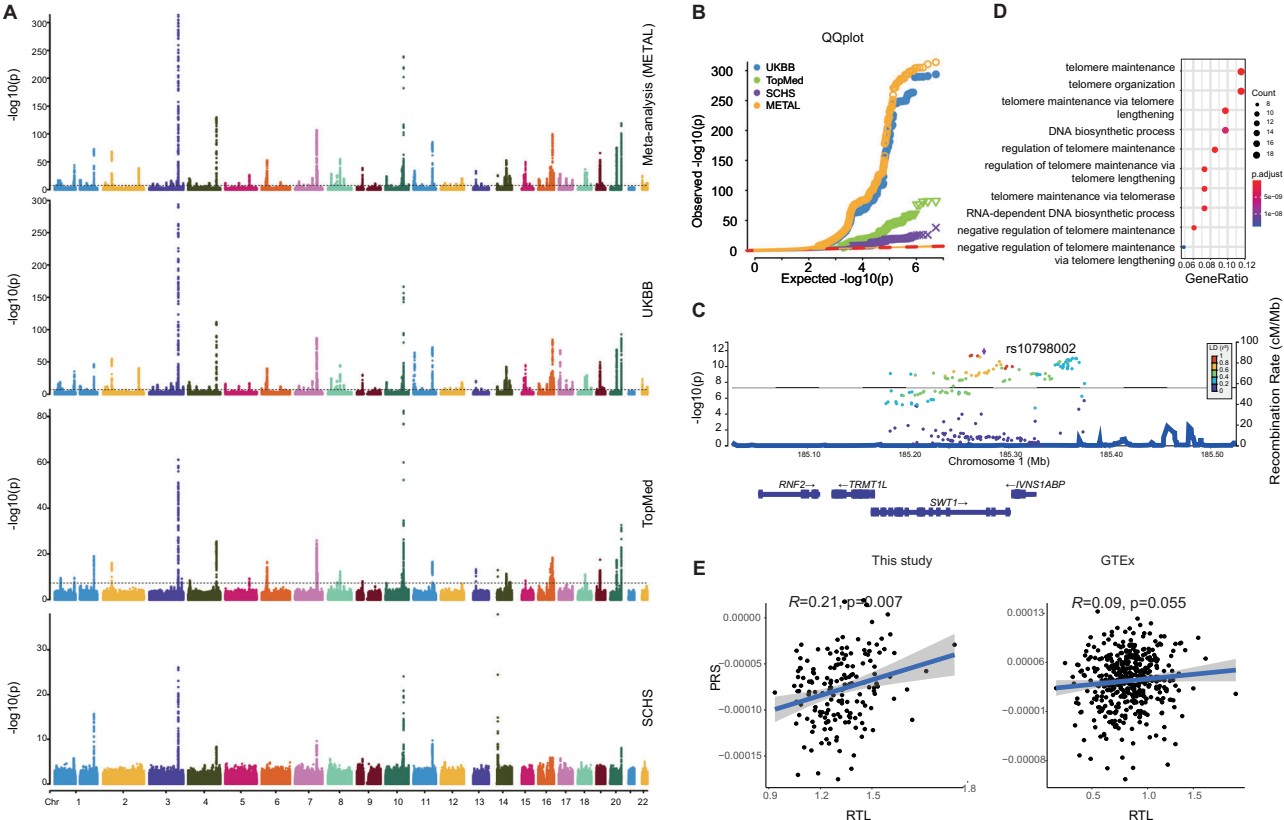

**Fig. 2 | Trans-ancestral TL GWAS and PRS analysis. A** Manhattan plot of various TL GWASs and GWAS meta-analysis. The x-axis represents the genome in physical order; the y-axis shows -log10 two-sided *P*-values for all variants using an inverse-variance weighted fixed effects model and a total sample size *n* = 644,553. **B** The quantile-quantile (Q-Q) plot compared the two-sided *P*-values generated from this fitted distribution against the observed *P* values. **C** LocusZoom plot for regional associations of a locus associating with TL in *SWT1* gene, SNPs are colored according to their LD with the lead SNP, rs10798002. The left y-axis shows association -log10 two-sided *P*-values for all SNPs in this locus, the right y-axis shows the recombination rate and the x-axis shows the chromosomal position. The bottom of plot shows the near genes. **D** Dot plot of GO enrichment for the nearest genes of TL-associated variants. The diameter indicates the number of genes overlapping the gene ontology term and the color indicates the BH-adjusted enrichment *P* value. **E** Scatter plot for RTL vs. PRSs with a simple linear regression line fitted in this study (*n* = 166) and GTEx (*n* = 442). The grey shade area represents 95% confidence interval.

---

showed that PRS is still significantly associated with TL (*r* = 0.18, *P*-value = 0.023, Supplementary Fig. 7). Collectively, combinatory analysis of trans-ancestral GWAS and placental TL measured by TRF assay indicated that human TL could be determined and predicted only genetically.

**Association of placental TL maintenance with telomere regulatory genes and functional gene connectivity**

Since 91.5% of sentinel variants of TL GWAS loci are located in the noncoding genomic region, investigating their regulatory potential on gene expression would improve the accuracy of genetic/transcriptomic-based TL predictions. Thus, we profiled transcriptomics using RNA-seq for the 166 placental samples with paired genotypes and TL measurements. Through rigorous literature review and database search, we collected genes postulated to influence telomere length regulation. Our curated sets encompass genes associated with the regulation of telomerase activity (81 genes), telomere capping (5 genes), and alternative lengthening of telomere mechanisms (ALT, 24 genes). These datasets now serve as a foundation for our ensuing analysis (Supplementary Data 7). We first evaluated the association between placental RTL and each of collected telomere regulatory genes with multiple comparison correction. As the results, for genes involved in telomerase activity regulation, *NFX1* (*r* = 0.24, *P* value = 0.002, false discovery rate (FDR) < 0.2) and *BMI1* (*r* = 0.22, *P* value = 0.004, FDR < 0.2), which regulated hTERT, were positively correlated with placental RTL (Fig. 3A). The expression levels of major telomerase

catalytic subunits in telomerase activity regulation pathway, including *TERT, DKC1, NOP10* and *WRAP53*, were not correlated with placental RTL (Supplementary Fig. 8), whereas *TERC* and *NHP2* expressions were undetectable in placenta (i.e., transcripts per million (TPM) = 0 in all samples). For genes involved in telomere capping, a SHELTERIN component *TPP1* showed a moderate correlation with placental RTL (*r* = 0.19, *P* value = 0.015, FDR > 0.2) (Fig. 3A), unlike other protein subunits of SHELTERIN and CTC1-STN1-TEN1 (CST) complexes, such as *TINF2, RTEL1, POT1*, and *CTC1* (Supplementary Fig. 8). *CGGBP1* was also moderate correlation with placental RTL (*r* = 0.18, *P* value = 0.021, FDR > 0.2). Interestingly, we observed that placental RTL was positively correlated with some components of the ALT pathway, including *ATRX* (*r* = 0.17, *P* value = 0.022, FDR > 0.2), *DAXX* (*r* = 0.21, *P* value = 0.008, FDR > 0.2), and *SMARCAL1* (*r* = 0.23, *P* value = 0.003, FDR > 0.2) (Fig. 3A and Supplementary Fig. 9). To mitigate the potential impact of infant sex and maternal age on our association test, we then applied multi-variable linear regression and t-tests to examine the relationships between RTL and gene expressions. Infant sex and maternal age were treated as covariates in this analysis. Post-adjustment, placental RTL exhibited a positive association with *NFX1* (*P* value = 0.006) and *SMARCAL1* (*P* value = 0.005) expressions (Supplementary Data 7). However, most of these correlations did not achieve statistical significance following adjustment for covariates and multiple testing corrections. This highlights the need for more extensive research, potentially involving a larger sample size, to further explore these associations.

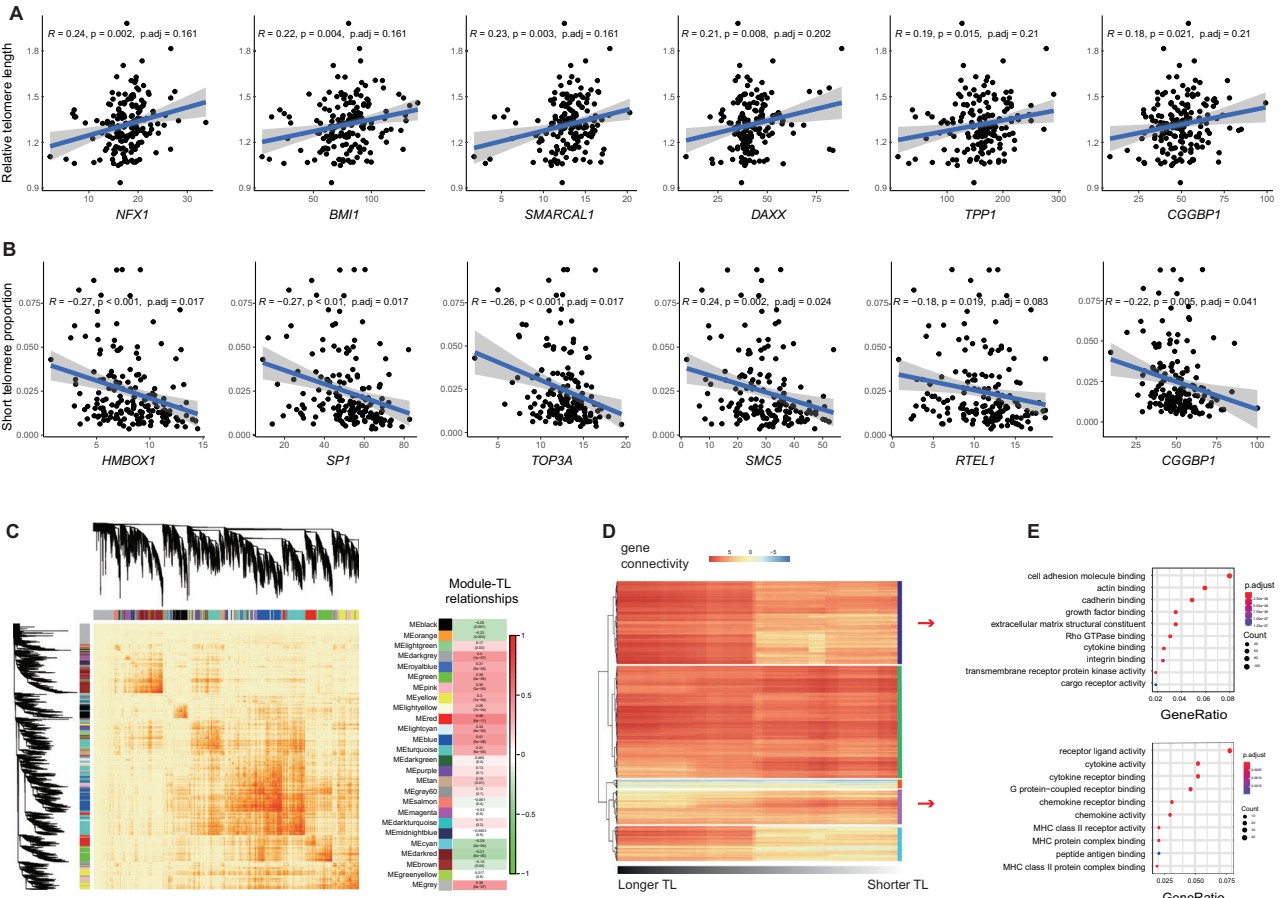

**Fig. 3 | Gene expression patterns of placental TL maintenance. A** Scatter plots shows the associations between RTL and top2 genes involved in each of telomerase activity, telomere capping, and ALT pathways. The grey shade area represents 95% confidence interval. P represents the two-sided *P*-value of Pearson's correlation test, p.adjust represents the BH-adjusted *P* values. **B** Scatter plots shows the associations between STP and top2 genes involved in telomerase activity, telomere capping, and ALT pathway. The grey shade area represents 95% confidence interval. The two-sided *P*-values of Pearson's correlation test were 3.00e-4 for *HMBOX1*, 3.37e-4 for *SP1*, 6.47e-4 for *TOP3A*. **C** Dendrogram of module eigengenes based on

dissimilarity measure (1-TOM) and the associations between each module and TL. **D** Heatmap of the connectives for high-variance genes based on about 25% of samples, while the left-most is based on samples with upper-quartile TLs, and the right-most is based on samples with lower-quartile TLs, lower score represents a low overlap and larger score represents a high overlap between the genes. **E** Dot plot of GO enrichment for genes clusters that predominately have higher connectives in samples with long TL (upper) or short TL (bottom). The diameter indicates the number of genes overlapping the gene ontology term and the color indicates the BH-adjusted enrichment *P*-value.

Moreover, we found that several canonical telomere maintenance genes were significantly correlated with placental STP (Supplementary Data 8). Notably, genes involved in telomerase activity regulation, such as *SP1* ($r = -0.27$, *P* value = 0.0003, FDR < 0.1), *HMBOX1* ($r = -0.28$, *P* value = 0.0003, FDR < 0.1), *EGFR* ($r = -0.26$, *P* value = 0.001, FDR < 0.1), exhibited significant associations with STP (Fig. 3B), whole some genes involved in telomere capping showed weak correlation (Supplementary Fig. 10). Furthermore, genes involved in ALT pathway, such as *TOP3A* ($r = -0.26$, *P* value = 0.001, FDR < 0.1), *SMC5* ($r = -0.24$, *P* value = 0.002, FDR < 0.1) were significantly associated with STP. This phenomenon suggested that short telomere phenotype could be an effective indicator sensitive to changes in telomere regulatory genes in the placenta.

To inspect the underlying biological functions associated with placental TL maintenance, we performed two gene module analyses based on transcriptome data from the 166 placentae. First, we used a weighted gene co-expression network analysis (WGCNA)[30] to construct the co-expression networks of the human placenta and identified 26 network modules (Fig. 3C). The correlation test between each module eigengene (ME) score and RTL identified turquoise module was significantly associated with RTL, and the genes in this module were enriched in the ubiquitin-like protein transferase activity

(GO:0019787, adjusted *P* value = 1.61e−06) (Fig. 3C and Supplementary Fig. 11); this finding was supported by previous studies on the ubiquitin-like proteins on telomere regulation[31,32]. Second, to investigate whether RTL affects gene connectivity in the placenta, we calculated the connectivities of 4679 genes with high variance from the 166 placental RNA-seq data, starting from the upper quarter RTL (left side of Fig. 3D) and subsequently added one sample with shorter RTL and removed one sample with longer RTL to recalculate the connectivities. Based on the hierarchical clustering of the RTL-driven gene connectivities, we detected five gene clusters by Elbow method. For example, 1424 genes in cluster 1 were predominantly interconnected among samples with longer RTL and these genes were significantly enriched in the cell adhesion molecule binding (GO:0050839, adjusted *P* value = 1.39e-20), while 585 genes in cluster 4 were highly linked among samples with shorter RTL and these genes were related to receptor ligand activity (GO:0048018, adjusted *P* value = 6.16e-09) (Fig. 3E).

**Integrating placental eQTL and TL GWAS for systematic prioritization of TL-causal genes**

Recent genetic studies have identified many new TL-associated genes by eQTL-based methods[11,33], but they mainly employed eQTLs derived

from adult tissues, probably leads to biased estimation through unobserved lifetime exposures. A genome-wide *cis*-eQTL mapping on 166 placental samples from Asia population (see methods for details) was conducted to examine new TL-causal genes based on our trans-ancestral GWAS meta-analysis. Compared to the previous placental eQTLs on European cohort[34], we observed similar distributions of genomic distances to the gene transcription start site (TSS) and end site (TES) (Supplementary Fig. 12A), but identified 3913 more eQTL-associated genes (eGenes) (Supplementary Fig. 12B). The effect sizes of eQTLs in the two cohorts were also correlated (Supplementary Fig. 12C, r = 0.42, P value < 2.2E-16). Based on RoadMap chromHMM 15 core states of placenta[35], we observed that most of the placental eQTLs are located in the active chromatin regions and significantly enriched in the placental active promoter (TssA) and enhancer (Enh and EnhG) states (Supplementary Fig. 12D, E). These results indicated the validity and the gained power of our placental eQTL mapping.

To systematically prioritize the potential TL-causal genes, we integrated our placental eQTLs and aforementioned trans-ancestral GWAS results using two complementary statistical strategies. First, colocalization analysis via COLOC[36] was performed to test the shared causal variants between gene expression (from eQTL) and TL trait (from GWASs). We provide results of the colocalization with strong evidence using a rigid standard (PP4 ≥ 0.8 and PP4/PP3 ≥ 5) and the likely colocalization with suggestive evidence using a liberal standard (PP4 ≥ 0.5 and PP4/PP3 ≥ 3). We identified 17 and 43 signals with strong and suggestive evidence of colocalization between placental eQTL and TL GWAS loci, respectively (Supplementary Data 9). Second, Fusion Transcriptome-wide association analysis (TWAS)[37] and Summary-based Mendelian Randomization (SMR)[38] were applied to test for a significant genetic correlation between *cis*-expression and GWAS signal. Thus, we observed 64 genes reaching transcriptome-wide significance (FDR < 0.1, two-tailed Z-test) in TWAS (Fig. 4A, Supplementary Data 10) and identified 61 genes showing a potential association with TL after heterogeneity in dependent instruments (HEIDI) test (FDR < 0.1) in SMR (Supplementary Data 11), respectively.

The intersection of gene prioritization results from COLOC with suggestive evidence and the union of TWAS and SMR retrieved 23 likely causal genes related to TL (Supplementary Data 12). These candidates encompassed several genes responsible for canonical telomere regulation, such as telomere length maintenance[39] and telomere end protection[40]. Moreover, some of the mechanisms underlying the causal genes have been explored recently. For example, *TSPYL5* is required to maintain POT1 protein levels and suppresses POT1 poly-ubiquitination and degradation exclusively in ALT cells[41]. *GEN1* is required for telomere replication and prevents the cutting of telomeres[42]. *RFWD3* plays a role in DNA damage response and facilitates translesion DNA synthesis[43]. *ATE1* encodes an arginyltransferase for ubiquitin-dependent degradation and is associated with sub-telomeric regulation[44]. We also discovered several new genes whose TL-related function was rarely documented, such as *RRM1*, *MMUT*, *KIAA1429*, *YWHAZ*, *PEX6*, *POLI*, *CDC25B*, and *HDDC2*, and exemplified strong evidence of positive causal associations between genetic determined expressions and TL in genomic loci of four new genes (Fig. 4B–I). Collectively, the stringent prioritization of TL-causal genes based on placental eQTL and large-scale TL GWAS summary information would provide new insight for understanding TL regulation.

## Experimental validations of top prioritized genes in TL regulation

To evaluate the causal effect of the new hits in our prioritization, we functionally verified the positive regulation of TL by perturbing the four genes screened above: *RRM1*, *MMUT*, *KIAA1429*, and *YWHAZ*. Briefly, HTR8/SVneo cell lines were established by immortalizing a physiological extravillous trophoblast cell by transfection with a plasmid containing the simian virus 40 large T antigen[45]. Stable HTR8/

SVneo cell lines were established by the knockdown of the above four genes by shRNA plasmids. Next, we detected whether TL in these cell lines would shorten via southern blots of TRF assays to verify the likely causal correlation between investigated genes and TL.

*RRM1* gene encodes the large and catalytic subunit of ribonucleotide reductase (RNR), an enzyme for converting ribonucleotides into deoxyribonucleotides, which is essential for DNA replication and DNA repair processes[46]. However, the direct correlation between human RNR and TL has not yet been established. Consequently, the knockdown of *RRM1* gene in eight monoclonal HTR8/SVneo lines consistently showed telomere shortening compared to control cells (Fig. 5A, B). One cell clone (shRRM1-2-6) was picked for continuous passage. The results demonstrated that TL was gradually shortened with cell passage (Fig. 5C), further confirming the positive regulatory effect of the *RRM1* gene on TL in the placenta. Besides, *MMUT* gene encodes the mitochondrial enzyme methylmalonyl-CoA mutase. In humans, the product of this gene is a vitamin B12-dependent enzyme that catalyzes the isomerization of methylmalonyl-CoA to succinyl-CoA, yet the causal correlation between *MMUT* and TL is unknown. We also found that *MMUT* knockdown in most monoclonal HTR8/SVneo lines significantly reduced TL (Fig. 5D, E). With continuous passage, the TL of the shMMUT-1-2 monoclonal cell line shortened continually (Fig. 5F). Finally, the inhibition of the other two new genes *KIAA1429* (a vital component of the m6A methyltransferase complex) and *YWHAZ* (tyrosine3-monooxygenase/tryptophan 5-monooxygenase activation protein zeta) showed a similar pattern of telomere shortening (Fig. 5G–J). This functional validation greatly supports the causal association between these enzymes and TL maintenance.

## Incorporating genetic and transcriptomic information for accurate TL prediction

This study, together with previous GWAS findings, has strengthened the polygenic basis of TL variation, yet the genetic determinants for TL explained by all genome-wide variants were not substantial (<10% variance explained)[10–12]. The accurate estimation of TL only relies on a single angle of information, such as genetic or epidemiological factors, which shows a low agreement with actual TL measured by different biochemical assays[16,47]. To improve the performance of TL prediction and facilitate TL-related clinical applications[48,49] when accurate TL measurement (such as TRF-based or fluorescent in situ hybridization (FISH)-based test) is absent, we incorporated transcriptomic information of placental tissue into TRF-based RTL prediction model and systematically compared it to the existing strategies. First, an elastic net regression model was constructed on our 166 placental multi-omics data and individual demographic information. Next, we inspected the ability of TL inference using transcriptional risk score (TRS)[50,51] over static PRS. As a result, the TRS model based on genes regulated by variants linked to TL ($r^2 = 0.48$, P value = 1.38e-20, 10-fold cross-validation) outperformed the PRS-based model ($r^2 = 0.10$, P value = 3.17e-05) in predicting placental RTL (Fig. 6A), suggesting that transcriptomic information reflects an additional layer of TL determinants than solely genetic information. Since both PRS- and TRS-based models largely depend on GWAS significant variants, we investigated whether incorporating expression signatures from a specific number of genes could enhance the performance of TL prediction. Thus, the elastic net regression was applied to model the TRF-based RTLs of 166 placentas alone with individual PRS and gender information, resulting in 32 selected genes, independent of PRS, which showed non-zero and significant coefficients (see Methods for details) and found that TL prediction model building on transcriptomic score calculated from the expression of these 32-gene signature (TS-32Gene) surpassed the TRS-based model ($r^2 = 0.48$, P value = 3.18e-24) (Fig. 6A). Additionally, the network enrichment analysis by EviNet[52] showed that both signature genes and TL-causal genes were enriched in DNA replication signaling

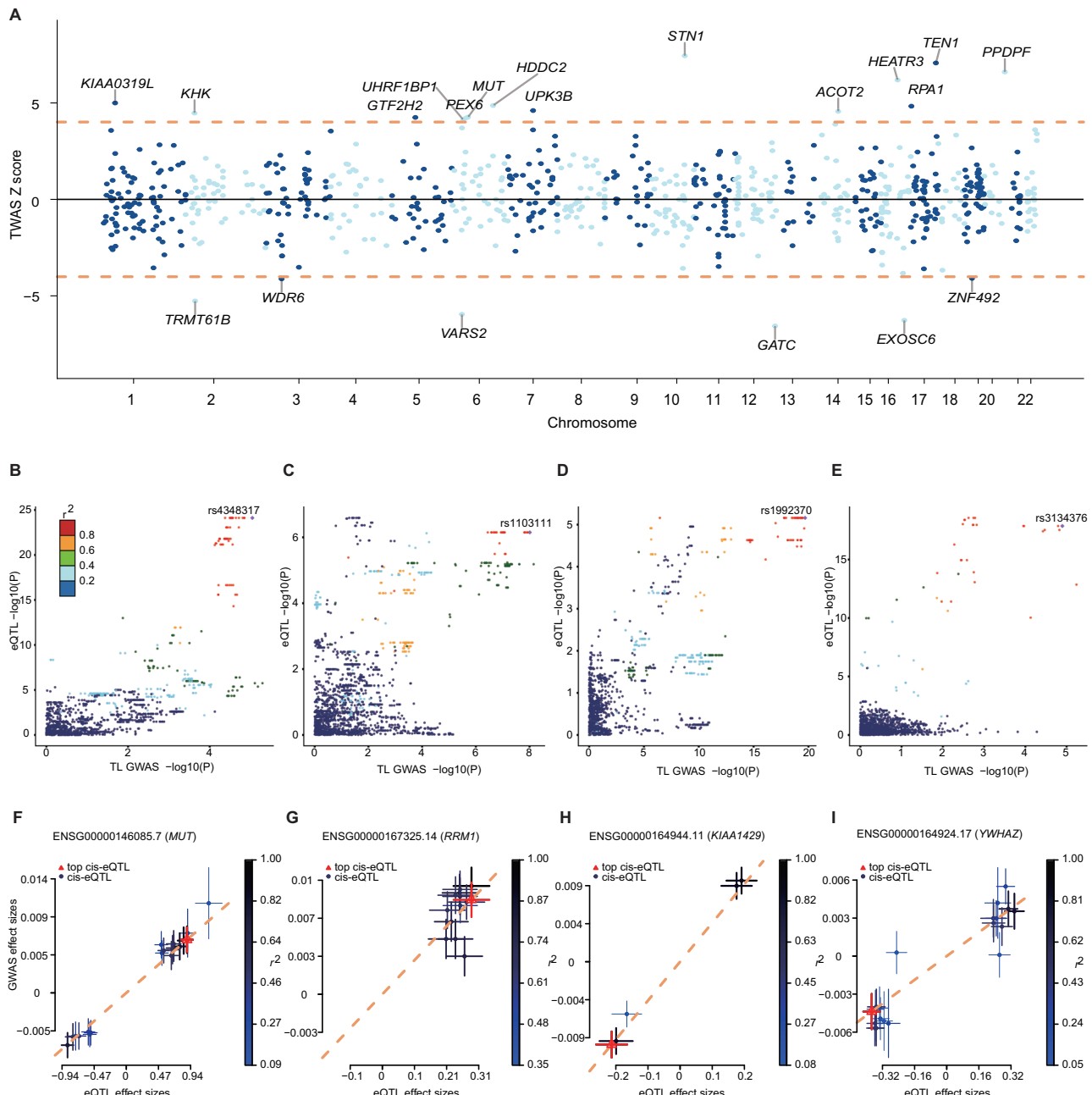

**Fig. 4 | Causal associations between genetically determined gene expressions and TL. A** Manhattan plot of transcriptome-wide association results. The x-axis represents the genome in physical order; the y-axis shows Z score for all genes. Genes that passed multiple testing corrections (FDR < 0.1) are highlighted in red and labeled with gene name. **B–E** LocusCompare plots for the (**B**) *MMUT*, (**C**) *RRM1*, (**D**) *KIAA1429*, and (E) *YWHAZ* loci, where the GWAS signals (x-axis) colocalized the eQTL signals (y-axis). LD is colored with respect to the GWAS lead SNPs. The lead SNPs are plotted as a purple upright triangle. The x-axis and y-axis show the -log *P* values for variants located at the respective loci. **F–I** Scatter plot of the effect sizes of variants reported in TL GWAS and placental eQTLs from (**F**) *MMUT*, (**G**) *RRM1*, (**H**) *KIAA1429*, and (**I**) *YWHAZ*. Effect sizes of the variants in the TL GWAS (y-axis) and eQTL (x-axis) are plotted. Error bars indicate 95% confidence interval. The red triangle shows the top *cis*-eQTL, blue circles indicate *cis*-eQTLs. Error bars show the standard errors of the SNP effects.

pathway (Supplementary Fig. 13), indicating that TS-32Gene represents a unique and powerful predictor of TL. To further gain predictive performance, we combined transcriptomic signature and genetic determinants of TL on several full models training. Notably, the combination model of TS-32Gene, PRS, and TRS exhibits the best performance ($r = 0.85$, $r^2 = 0.72$, *P* value = 1.70e-47) among all trained models (Fig. 6A, B).

To evaluate the validity of TRF-based RTL prediction model independently, we applied the most practical model (with parsimonious information) only based on TS-32Gene and PRS to GTEx multi-omics data and observed a good agreement between the predicted and observed RTL measured by Luminex-based assay across different GTEx tissues (Fig. 6C, $r = 0.26$, *P* value = 5.92e-45), especially in whole blood, ovary, and esophagus tissues (Supplementary Data 13). This result not only demonstrated the generalizability of our TL prediction model in different contexts but also implied a shared pattern of TL regulation between the placenta and other tissues. In addition, TL estimated by TelSeq[53], a sequencing-based TL measurement (see Methods for details), showed a weak correlation with Luminex-derived TL in GTEx whole blood tissues (Fig. 6D, $r = 0.13$, *P* value = 0.006), further indicating the superiority of our strategy. Further, to substantiate our TL prediction strategies in an additional cohort, we

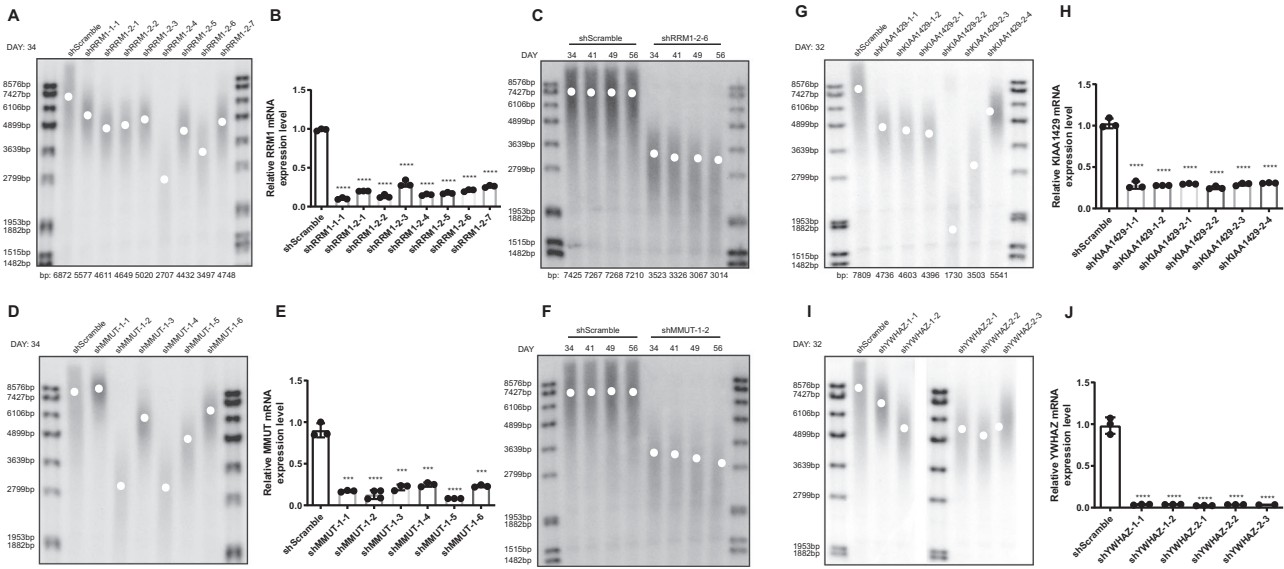

**Fig. 5 | Experimental validation of TL regulation by perturbing expression of *MMUT, RRM1, KIAA1429, and YWHAZ*.** **A** 34 days after HTR8/SVneo cells were infected with shRRM1 lentiviral particles, 8 shRRM1 monoclonal cell lines grown in 96-well plates were transferred to 6-well plates. TRF assay was used to measure the TL of these cell lines. **B** *RRM1* RNA levels were estimated by qPCR and analyzed by GraphPad Prism software version 6.0. Data are represented as mean ± SD (*n* = 3). The two-sided *P* values of t test were (1.36e-7, 7.23e-8, 6.28e-7, 1.01e-5, 1.35e-7, 1.48e-7, 3.20e-7, 4.08e-7). ****P* value < 0.0001. (**C**) HTR8/SVneo cells stably expressing control (shScramble) and shRRM1 were passaged over time (DAY) and examined for average TL by TRF (*n* = 4). **D** 34 days after HTR8/SVneo cells were infected with shMMUT lentiviral particles, six shMMUT monoclonal cell lines grown in 96-well plates were transferred to 6-well plates. TL of these six cell lines was measured by TRF assay. **E** *MMUT* RNA levels were estimated by qPCR and analyzed by GraphPad Prism software version 6.0. Data are represented as mean ± SD (*n* = 3). The two-sided *P*-values of t test were (1.23e-04, 2.34e-05, 2.05e-04, 2.19e-04, 7.63e-05, 1.78e-04). ***P* value < 0.001; ****P* value < 0.0001. **F** HTR8/SVneo cells stably expressing control (shScramble) and shRNA sequences against MMUT were passaged over time (DAY) and examined for average TL by TRF (*n* = 4). **G** 32 days after infection of shKIAA1429 lentiviral particles, six shKIAA1429 monoclonal cell lines from 96-well plates were grown in 6-well plates. The TL of these cell lines was measured by TRF assay. (**H**) *KIAA1429* RNA levels were tested by qPCR and analyzed by GraphPad Prism software version 6.0. Data are represented as mean ± SD (*n* = 3). The two-sided *P* values of t test were (6.15e-05, 2.51e-05, 2.82e-05, 2.59e-05, 3.14e-05, 2.96e-05). ****P* value < 0.0001. **I** 32 days after infection of shYWHAZ lentiviral particles, 5 shYWHAZ monoclonal cell lines grew from 96-well plates to 6-well plates. TL of these cell lines was measured by TRF assay. **J** *YWHAZ* RNA levels were estimated by qPCR and analyzed by GraphPad Prism software version 6.0. Data represent mean ± SD (*n* = 3). The two-sided *P*-values of t test were (7.80e-05, 7.80e-05, 7.30e-05, 7.70e-05, 1.01e-03). ****P*-value < 0.0001.

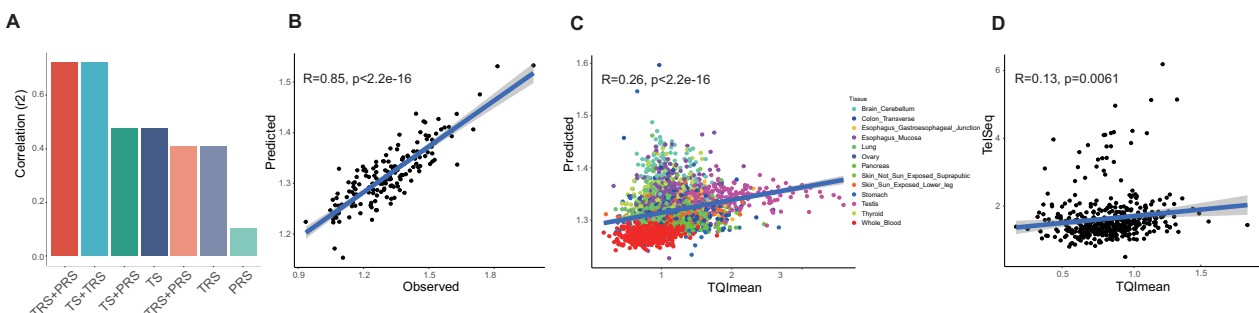

**Fig. 6 | Performance of placental TL prediction model.** **A** Bar chart for performance (*r²*) of elastic net models based on different feature combinations. **B** Scatter plot shows the actual TL values in placenta against the values predicted by the best model (TS + TRS + PRS). The grey shade area represents 95% confidence interval. Two-sided *P*-value was computed for Pearson's correlation test. **C** Scatter plot shows the actual TL values in GTEx tissues against those predicted by the model, only GTEx tissues with >100 samples were used for validation. The grey shade area represents 95% confidence interval. Two-sided *P* value was computed for Pearson's correlation test. Dots are colored according to their tissue types. **D** Scatter plot shows the correlations between TelSeq and TQImean for GTEx whole blood individuals with a simple linear regression line fitted. The grey shade area represents 95% confidence interval. Two-sided *P* value was computed for Pearson's correlation test.

exploited the UKBB Chinese dataset, encompassing both genotype and telomere length data. By leveraging estimated gene expressions from genotype and placental eQTL, the observed correlation between predicted and actual TL values was significant (Supplementary Fig. 14, r = 0.35, *P* value < 2.2e-16). This underscores the robustness and precision of our approach that amalgamates both PRS and transcriptomic data for TL prediction.

## Discussion

Telomere shortening is a classical hallmark of cell senescence and aging, and there are many known elements that contribute to the individual variations of TL, such as genetic, environmental, and lifestyle factors[2]. Although several large-scale GWASs have identified a large number of genetic loci associated with TL,[11,12,33] however, the true causal genes underlying the telomere content regulation are yet to be

elucidated. By leveraging TRF assay, genotyping chip, and RNA-seq on hundreds of placenta samples together with trans-ancestral TL GWAS and functional validations, we systematically investigated the causal genes and developed a powerful prediction model for human TL.

A recent study by GTEx consortium has investigated the determinants of TL across various adult human tissues and cell types[16]. However, samples included in GTEx experienced life-course exposures to the external environment and physiological status. The inherently short or long TL might be largely determined at birth and may be crucial for lifelong health[28]. Studies on TL of human tissues with a primitive state are lacking. The placenta embeds in the maternal uterus, allowing nutrition delivery and promoting the growth and development of the fetus[54]. In addition, heritability estimates on adults are often affected by environmental factors, impeding the identification of the true genetic determinants. Since newborn TL could predict later life TL[55], it is critical to investigate TL determinants of the newborn. However, no existing study has harnessed a significant number of early samples without postnatal environmental exposure to study TL.

In the placenta, the telomere is less affected by extraplacental exposure or other non-genetic effects, which provide an ideal proxy for studying the genetic determinants and causal genes of TL. Using the TRF measurement, we are able to not only calculate the mean TL, but also characterize the STP. Next, this study investigated the correlation between RTL and genes involved in telomerase activity regulation, telomere capping, and the ALT pathway. We found that placental RTL and STP were significantly correlated with genes involved in the telomerase activity regulation and ALT pathway, but not with telomerase subunit genes. Despite our data indicating an association of increased *ATRX, DAXX*, and *SMARCAL1* expression with longer placental telomere length, this contradicts conventional ALT mechanisms seen in cancer[56], where such genes are typically lost. This discrepancy questions ALT discussions' applicability in healthy placenta, underscoring the need for larger-scale studies. Besides, we observed a discrepancy in RTL characteristics between the placenta and other adult human tissues. Consistent with previous studies, the present study revealed that placental RTL exhibits intra-tissue homogeneity but is longer than other somatic adult tissues[23,24,57]. Nevertheless, the placental RTL of male infants is slightly longer than that of female infants, while RTL is longer in females than males in adult tissues, suggesting that the females may sustain long RTL during postnatal developments. Furthermore, recent studies have reported that longer telomere length is associated with an increased risk of adolescent-onset ependymoma and osteosarcoma[58,59]. These findings highlight the criticality of investigating TL regulation during early life stages and pave the way for future research exploring how genetic effects, that may alter TL, could possibly contribute to developmental disorders and diseases.

Studies from multiple worldwide cohorts were pooled to perform trans-ancestral GWAS meta-analysis in over 500,000 individuals. The power to detect genome-wide significant signals associated with TL was improved; as a result, we could detect 20 new genetic associations and recover 87% of the previously reported TL GWAS significant loci. Notably, PRSs of 166 placental samples constructed by TL-associating variants identified via trans-ancestral GWAS were significantly associated with RTL. However, when evaluating PRSs using either GWAS variants from a single cohort or genotypes from GTEx whole blood/ UKBB leukocyte samples, we only detected weaker RTL correlations, suggesting that leveraging the trans-ancestral GWAS and TRF-based TL measurement could boost the predictive power of PRS on TL. On the other hand, gene expressions in placental tissue are less perturbed through lifetime exposures. Based on genotype and transcriptome profiling from 166 placental samples, we conducted genome-wide *cis*-eQTL mapping and performed TL-causal gene discovery together with our trans-ancestral GWAS results. Complementary statistical methods (such as COLOC, Fusion-TWAS, and SMR) yield 23 likely causal genes

related to TL, and some are rarely or inadequately associated with TL, such as *MMUT, RRM1, KIAA1429*, and *YWHAZ*, which showed a positive regulation of TL. By establishing HTR8/SVneo cell lines with a stable knockdown of these four genes, we validated their functional relevance in maintaining TL via TRF assay. However, the biological mechanisms and tissue/cell-type specificity of these new TL-causal genes still need an in-depth investigation in the future. Supposedly, some causal genes may not have been detected in this study due to the limitations of sample size, population difference, and tissue specificity of TL maintenance and regulation.

Recent genetic studies have made significant strides in identifying new genes associated with TL using eQTL-based methods. However, these studies have primarily relied on eQTLs derived from adult tissues, potentially resulting in biased estimations due to unobserved lifetime exposures. In this manuscript, we present our findings on TL-associated genes, highlighting our unique approach that addresses these limitations. By incorporating tissue-specific sources, as well as integrating transcriptome, sequence variants, and TL data, we aim to provide a comprehensive understanding of TL regulation. While our study showcases its unique aspects, it is essential to highlight the potential drawbacks, such as the absence of additional functional genomic annotations and predictions utilized in recent adult tissue-based eQTL studies, as well as incomplete causal gene prioritization strategies[60,61]. Our study acknowledges the limitation of insufficient sample size, which can influence the detection of associations between genetic variants and gene expression levels in eQTL analysis. Future investigations with larger sample sizes are anticipated to enhance statistical power, enable the detection of weaker eQTL signals, and improve the precision and generalizability of the results. This advancement is expected to provide more reliable and comprehensive insights into the genetic regulation of gene expression in the placenta. Furthermore, the accuracy of meta-analysis can be attenuated in the presence of cross-study heterogeneity, which can be attributed to several factors. One significant consideration is the meta-analysis of multiple TL studies, which might utilize slightly different definitions of the phenotype under investigation. Consequently, the effect sizes across these studies may vary. Additionally, the presence of ancestry-specific effects can further contribute to the observed heterogeneity. Together, these factors highlight the importance of carefully interpreting GWAS results in the context of diverse study populations and varying definitions of phenotypes, thereby emphasizing the need for rigorous validation and comprehensive understanding of the underlying genetic associations of TL.

Although TRF assay is a gold standard to quantify TL, it also has many drawbacks; it is time-consuming, less cost-effective, and requires large DNA material[62]. Since the fine-grained determinants of placental TL could be explored effectively using both genetic and transcriptomic information, we constructed several TL prediction models and systematically compared them to the existing strategies. The proposed model relying on transcriptomic signature not only exhibited a great performance in our TRF-based data but also generalized well in the GTEx data. Our model performs better than a whole-genome sequencing (WGS)-based TL estimation method, TelSeq. The gained performance in the independent datasets, especially in whole blood, lung, and esophagus tissues, highlighted that the regulators of placental TL might have similar biological roles in other tissues. Thus, we speculated that our TL prediction strategy could assist the TL-related clinical applications[48,49] when accurate TL measurement (such as TRF-based or FISH-based test) is not reachable.

## Methods

### Sample collection and processing

The healthy singleton Chinese pregnancies ($n = 166$) were recruited prior to delivery at Tianjin Central Hospital of Gynecology Obstetrics, China. These study participants did not have any recorded medical

disorders or adverse pregnancy outcomes. The hospital ethics committees approved the collection and use of human placental samples (approval no. 2022KY071). All participants provided written informed consent before sample collection. Placentas were treated within 10 min of a vaginal delivery from full-term pregnancies (37 + 0–41 + 6 weeks). The average age of the participants was 32 ± 4.0 years, and no tobacco-smoking or alcohol-drinking behavior was noted. The cohort comprised 73 female and 93 male infants. No significant differences were observed in maternal age, maternal BMI, infant weight, and gestation weeks with regard to infant gender (P value > 0.05).

We optimized the placenta sampling protocol recommended in the Amsterdam Placenta Workshop Group Consensus Statement to allow standardized tissue collection for this study[63]. Specifically, four equidistant sampling points were selected from a hypothetical concentric area with a radius of 2 cm centered at the placental umbilical cord insertion point to collect a 1.5 × 1.5 cm full-thickness placental biopsy. The placenta was placed with the fetal side up and oriented with the largest umbilical artery on the fetal side of the placenta as a reference. To avoid contamination with cells of non-target origin, the membranes were excised and excess blood was removed using sterile filter paper. A processed full-thickness biopsy was divided into three equal parts, and samples were taken from both sides, located in the fetal and maternal layers. A total of eight samples were obtained per placenta. All samples were stored in RNAlater at −80 °C until extraction. To analyze the potential intraplacental variation of TL, one random sample from the four fetal layer-derived biopsies collected from each of the 166 placentas, was selected for subsequent testing, including TRF assay, genotyping, and RNA-seq.

This study investigated telomere length in healthy individuals using 231 blood samples and 12 skin samples (approval no. 2023YS105). The blood samples comprised 143 males and 88 females, with an average age of 46.9 years (ranging from 5 to 89 years). Peripheral blood samples of 0.5–2.5 ml were collected from the elbow vein of each participant and preserved in EDTA-coated venous blood collection tubes for subsequent genomic DNA extraction. Additionally, 12 skin samples were obtained from healthy individuals (6 males and 6 females) with an average age of 37.6 years (ranging from 26 to 59 years), and genomic DNA was extracted for further telomere length analysis. Furthermore, the study included normal adult cardiac and lung tissues (N = 6 each) for telomere length analysis. Ethical regulations were strictly followed, and the use of fetal samples was approved under protocol number 2022ky071-1. Approximately 100 mg tissue samples were used for genomic DNA extraction.

### TRF length analysis

TRF method combined with pulsed-field electrophoresis was used to detect the TL length of 166 placental tissues. Genomic DNA isolation kit (Biomiga, BW-GD2211-02) was used to extract genomic DNA from placental tissue and HTR8/SVneo cells in the following experiments. TRF length analysis was applied to measure the TL of these DNA samples. Briefly, 1 μg genomic DNA of each sample was digested with *HinfI* and *RsaI* and then analyzed by agarose gel electrophoresis in 0.5× TBE. Pulsed-field gels [1% (wt/vol)] were run at 6 V, 14 °C for 16 h, and normal electrophoresis gels [0.8% (wt/vol)] were run at 100 V, 0–4 °C for 3 h. Subsequent procedures, such as gel depurination, gel denaturation, gel neutralization, DNA transfer to membrane, hybridization with DIG-labeled telomere probe, chemiluminescent detection, and TRF length analysis, were conducted. The sequence of telomere probe is TAACCCTAACCCTAACCCTAACCC. As an internal control, HeLa cell DNA was added in the first lane of each experiment to correct for batch-to-batch variation, and the RTL of 166 placental tissues was estimated by the ratio of their TL *vs.* HeLa DNA's TL. Besides, we also consider the telomere <5000 bp as the short telomere and the ratio of the band's intensity <5000 bp to the total intensity of the entire telomere band as the STP. The quantification and normalization of TRF

length were performed using TeloTool[25]. To enhance the credibility of our data analysis, TeloTool was utilized with a stringent Fit threshold setting of 60%.

### Genotyping, imputation, and quality control

Placental samples were genotyped using the Asian Screening Array (ASA) 750k platform, an Illumina whole-genome single nucleotide polymorphism (SNP) chip designed based on a large-scale East Asian whole-genome sequencing data that encompasses about 750,000 markers. Genotype calling by ASA resulted in a dataset of 166 individuals typed at 738,980 markers. Data cleaning was performed using PLINK v1.9[64]. All genotyped variants were subjected to quality control (QC) before imputation. Consequently, variants with (1) call rate (<95%) in all samples, (2) MAF < 0.0001, and (3) departures from Hardy-Weinberg equilibrium (P value < 1E-5) were removed. Also, individuals with the following criteria were removed: (1) overall SNP genotyping call rate <95% and heterozygosity rate > 3 SD; (2) genetically inferred sex mismatches between genotype and self-report; (3) related individuals with an identity-by-descent value > 0.1875; Before imputation, we removed SNPs with C/G and A/T alleles to avoid strand flipping. Then, we used Michigan Imputation Server[65] to impute untyped SNPs by borrowing the LD information from all samples using GAsP with Minimac4 for imputation and Eagle v2.4 for phasing. Following imputation, any imputed variant with an imputation quality score <0.3 or MAF < 0.01 was removed.

To determine if DNA for genotyping included maternal side tissue, we employed the following steps using B allele frequency (BAF) and log R ratio (LRR)[66–68]. Firstly, we calculated the BAF and LRR values and plotted them on scatter plots or histograms to visually assess their distribution patterns. In nonmixed DNA samples, BAF values typically cluster around specific patterns based on the genotype (homozygous or heterozygous). Deviations from these expected patterns may suggest a DNA mixture. Subsequently, we identified BAF values that significantly deviated from the expected patterns, such as clustering around 0.5 for heterozygous SNPs. Substantial variations in BAF values could suggest the presence of DNA from multiple individuals. Additionally, we examined LRR values for aberrations that deviated from the normal diploid states. Copy number alterations or imbalances in LRR values might indicate the presence of a DNA mixture from multiple individuals. Finally, samples that did not pass the examination based on BAF and LRR criteria were excluded from further analysis.

### Transcriptome profiling and quantification

During DNA genotyping and TL measurement, we performed RNA-seq to quantify the genome-wide mRNA expression for 166 placental samples. An equivalent of 3 μg RNA per sample was used as an input material for the RNA sample preparations. RNA-seq was performed using the Illumina NEBNext® Ultra™ RNA sample preparation protocol. The final libraries were sequenced on HiSeq 4000 platform using 150 bp paired-end chemistry and were run with a coverage goal of 80 M reads. Reads containing adapter and ploy-N and those with low quality were removed from raw sequencing reads using fastp[69]. Sequencing QC was used to obtain the overall quality, GC content, and adapter contamination using FastQC (http://www.bioinformatics.babraham.ac.uk/projects/fastqc). Then, we used STAR v2.5.3a[70] to align the paired-end reads to the reference genome. Gene annotation file was downloaded from GENCODE release 26 (https://www.gencodegenes.org/human/release_26.html). RNA-SeqQC v1.1.9[71] was applied to count the read numbers mapped to each gene. The genes were selected based on the following expression thresholds: ≥ 0.1 TPM and ≥ 6 reads count in at least 20% of samples.

### Co-expression network construction

We used WGCNA[30] to construct the co-expression modules and calculate the gene connectivities. The co-expression networks were

constructed with the soft power at 9, while other parameters were set at default. The adjacency was transformed into a topological overlap matrix (TOM), and the average linkage hierarchical clustering was performed according to the TOM-based dissimilarity measure. The module eigengene (ME) was the first principal component of a given module and could be considered a representative of the module's gene expression profile. The 26 MEs for the 26 distinct modules were each tested for the correlations with RTL. Then, gene connectivities were determined by calculating the connectivity values using soft-Connectivity function of WGCNA. Briefly, connectivity describes how strongly a gene is connected to all the other genes in the network. The absolute value of Pearson's correlation coefficient was calculated for all pairwise comparisons of gene expression values across all samples. The Pearson's correlation matrix was then transformed into an adjacency matrix. softConnectivity constructed the adjacency matrix and calculated the connectivity of each gene, i.e., the sum of the adjacency to the other genes. A total of 4679 genes exhibiting the top 30% high expression variance (captures more valid information) were selected for the co-expression analysis. K-means clustering was used to determine the gene groups for the connectivity shift. We also determined the number of clusters by Elbow method. Finally, GO enrichment analysis was carried out via R package clusterProfiler[72].

## Correlation test for TL and TL-related genes

By conducting a comprehensive literature review and keyword search on the Gene Cards website (https://www.genecards.org/) using terms such as "telomerase activity", "telomere capping", and "alternative lengthening of telomere", we meticulously compiled three distinct datasets. These datasets contain genes that potentially regulate telomere length, including genes related to telomerase activity regulation, telomere capping, and alternative lengthening of telomere for further analysis. Subsequently, we computed the Pearson correlation coefficient and evaluated its significance using a two-tailed t-test. To control the false discovery rate (FDR), we employed the Benjamini-Hochberg (BH) procedure, setting a significance threshold of <0.2 to identify statistically significant results after adjusting for multiple comparisons. Furthermore, we conducted a multivariable linear regression model, considering maternal age and infant gender as covariates in the analysis.

## eQTL mapping

The expression values for each gene were further inverse normal transformed across samples by trimmed mean of M-values[73]. eQTL mapping was performed using tensorQTL[74], a GPU-based method with high efficiency. Next, we used a linear regression model, with top 5 genotype principal components (PCs), age, and 30 PEER factors adjusted. Genotype PCs were computed based on the post-QC genotyping VCF using EIGENSTRAT[75]. To detect *cis*-eQTLs effects, we tested the nominal associations between all variant-gene pairs within a ± 1 Mb window around the TSS of each gene and estimated the beta-approximated empirical *P*-values to obtain appropriate significance thresholds based on 10,000 permutations of each gene. Multiple testing corrections were assessed using the Benjamini–Hochberg algorithm, with FDR across all *cis*-eQTL tests within each chromosome estimated. The placental chromatin state regions predicted by chromHMM[35] 15-core state model were downloaded from https://egg2. wustl.edu/roadmap/. We performed Fisher's exact test to investigate whether eQTLs were prone to be located in a specific chromatin state than expected. A two-sided *P*-value and odds ratio were calculated to measure the enrichment of eQTLs in the chromatin state regions.

## GWAS meta-analysis

We collected genome-wide summary statistics from three TL trait GWASs based on large-scale individuals, including SCHS[10], NHLBI TOPMed[12], and UK Biobank (UKBB)[11]. First, the variants with MAF < 0.01

were excluded. The rationale behind selecting this MAF cut-off value stems from the limitation of available GWAS summary statistics for the Singapore cohort, which only covered common variants (MAF ≥ 0.01). Thus, to ensure compatibility with the available data, we use the filter threshold at ≥1% for MAF across separate GWAS cohorts. We specifically excluded low-frequency variants to ensure the consistency of variant selection across the multiple GWAS studies we incorporated. Then a fixed-effect meta-analysis weighted by sample size of each study was conducted using METAL[76]. Genome-wide statistical significance for the meta-analysis was set at *P* value < 5E-8, HETEROGENEITY mode was set to determine whether the observed effect sizes were heterogeneous across samples. To recognize the new genetic loci in trans-ancestral meta-analyses, we first identified the associated genetic loci in TOPMed, UKBB, and SCHS meta-analyses at a threshold of *P* value < 5E-8. A locus was defined new in a trans-ancestral meta-analyses if it did not overlap with any loci of GWAS from a single cohort (R2 > 0.01 with any of reported TL loci). Manhattan and Q-Q plots were generated by CMplot[77]. In order to identify the positions of loci containing TL-associated variants, linkage disequilibrium (LD) clumping was conducted using PLINK v1.9. The variants were pruned with the following parameters: a *P*-value cutoff of 5E-8, at a genomic distance of 10 Mb, and $R^2$ < 0.001 with the lead SNP, using the LD structure of the 1000 Genomes Project as a reference panel. The HBB variants were removed from the results due to potential technical artifacts reported by UKBB.

## PRS analysis

Polygenic scores of TL were constructed using PRSice-2[78] to gauge the associations between reported variations of TL in general populations and in the current study. The scores were computed as the weighted sum of effect allele dosages, as a matrix multiplication of SNP dosages per individual by betas per SNP, i.e., the outcome is a single score of each individual's genetic loading for TL. Our measure of predictive power is the incremental $R^2$ from adding the score to a regression of the phenotype while adjusting for top five genotyping PCs, sex, and maternal age. The PRS was calculated by summing over all SNPs meeting a set of thresholds, respectively. We used the default *P*-value thresholds in PRSice-2 (from 5E-8 to 0.5, step size: 5E-5). All SNPs that met the specified threshold underwent LD pruning to reduce overfitting, utilizing the default settings (distance for clumping: 250 kb, $R^2$ threshold: 0.1). The null *P* value of the association of the best-fit GWAS *P* value threshold was converted to the empirical *P* value under 10,000 permutations. Pearson's correlations between PRS and RTL were used to compare the PRS analytical performance for the Chinese samples in UKBB, all samples in GTEx, and all samples in this study.

## Colocalization analysis

COLOC was applied to colocalize eQTL and TL signals which provided evidence of a putative causal correlation between the eQTL target gene and TL[36]. Herein, we used coloc.abf function implemented in the R package COLOC to perform colocalization analysis. We provide results of colocalization with strong evidence and likely colocalization with suggestive evidence using both a rigid standard (PP4 ≥ 0.8 and PP4/ PP3 ≥ 5) and a liberal standard (PP4 ≥ 0.5 and PP4/PP3 ≥ 3). The regional plot was generated using locuszoom (http://locuscompare.com/)[79], and LD was calculated based on the genotype of all individuals from 1000 Genomes project phase3[80].

## TWAS analysis

The summary-based TWAS was applied to GWAS meta-analysis data using FUSION following the pipeline described on their website (http:// gusevlab.org/projects/fusion)[37]. FUSION estimated the heritability of gene expression levels explained by SNPs in *cis* regions to each gene using the mixed-linear model (for instance, BLUP, BSLMM, LASSO, Elastic Net, and Top1 models). The weights for gene expression in the

placenta were calculated based on the correlation between SNPs and the placental gene expression while accounting for LD among SNPs. The genes that failed the heritability check (heritability $P$-value > 0.01) were excluded from further analyses. We restricted the $cis$-locus to 500 kb on the either side of the gene boundary. Then, the associations between the predicted expression of genes and TL were identified by FUSIOIN at default settings. Finally, the proportion of variance in gene expression, $P$-value, and Z-score was obtained from FUSION. TWAS Manhattan plot was generated using TWAS-plotter (https://github.com/opain/TWAS-plotter). TRS was constructed by the genetic value weighted by their Z-score in the TWAS.

## SMR analysis

The summary-based SMR method allowed us to infer the causal association between genetically determined gene expression and TL. The SMR test was developed to test the association between the exposure and the outcome using a single genetic variant as the instrumental variable[38]. Based on the assumption of SMR, SNPs are required to affect the TL only through the effects on gene expression. $cis$-eQTLs were used as the instrumental variables in this analysis. The HEIDI test was carried out to test the existence of linkage in the observed association. Rejecting the null hypothesis (i.e., $P_{HEIDI} < 0.01$) indicated the presence of two or more variants in high LD underlying the association. Thus, we used the default settings in SMR (i.e., MAF ≥ 0.01, excluded SNPs with LD $R^2$ between top-SNP > 0.90 and <0.05 and one of each pair of the remaining SNPs with LD $R^2$ > 0.90), and leveraged the FDR for multiple testing corrections.

## Elastic net regression

We also used an elastic net regression model to regress TL on maternal age, infant sex, TL PRS, TRS, and gene expressions. It is a regularized regression method that linearly combines the L1 and L2 penalties of the LASSO and ridge methods[81], emphasizing model sparsity while appropriately balancing the contributions of co-expressed genes. The raw values of all the features are standardized by removing the mean and scaling to the unit variance before training. Optimal regularization parameters were estimated via 10-fold cross-validation. The alpha parameter was set to 0.14, and the lambda value from the best prediction model selected by exhaustive grid search was set to 0.18. The elastic net regression model automatically selected features for building a TL predictor and reported an effect size for each feature. To compare the incremental predictive power of PRS and TRS, we also trained two models that included maternal age, infant sex, and TL PRS only or TRS only. Since our sample size was <200, we did not leave a testing set for validation but used a 10-fold cross-validation strategy. Instead, GTEx was utilized as an independent validation source, where only GTEx tissues with > 100 samples are used for validation. The correlation ($r^2$) between the predicted and the true TL across all samples was used to evaluate the accuracy. Moreover, we compared the performance of the elastic net regression model used in this study and a WGS-based TL estimation tool, TelSeq[53]. The TelSeq tool estimates the average TL using counts of sequencing reads containing a fixed number of telomere signature TTAGGG repeats. A repeat number of 12 and a GC content window of 48–52% was applied in this calculation. TelSeq was also used to estimate the TL for 670 GTEx whole blood samples based on sequence alignment files derived from WGS data, while $r^2$ between the predicted and the true TL across all samples was used to compare the performance. To further validate our TL prediction strategies in UKBB Chinese data, which provides both genotype and telomere length information. We employed a strategy akin to the one used in TWAS. Specifically, we leveraged the weights obtained from TL eQTLs to predict gene expression from the available genotype data in the UKBB Chinese dataset using FUSION. Then the elastic net regression model was trained on age, sex, PRS, and predicted gene expressions.

## Cell culture

HEK 293 T cells were grown in DMEM (Corning, USA) supplemented with 10% fetal bovine serum (LONSERA, UY) and 1% penicillin-streptomycin. HTR8/SVneo cells were cultured in 1640 (Corning) medium containing 10% fetal bovine serum and 1% penicillin-streptomycin. The optimal culture conditions in the incubator were 37 °C, 5% $CO_2$, and humidity of about 95%.

## shRNA design and plasmid construction

shRNA sequences were introduced into pLKO.1-puro vectors. The targeting sequences for various shRNAs oligos are as follows:

shMMUT-1: 5'-CCCTTGTATTCCAAGAGAGAT-3';
shRRM1-1: 5''CCCACAACTTTCTAGCTGTTT-3';
shRRM1-2: 5'-GCTGTCTCTAACTTGCACAAA-3';
shKIAA1429-1: 5'-CGGAATATGAAGCAACAAATT-3';
shKIAA1429-2: 5'-CGCTGAGCAAAGTTCTCATAT-3';
shYWHAZ-1: 5'-GCAGAGAGCAAAGTCTTCTAT-3';
shYWHAZ-2: 5'-GCAATTACTGAGAGACAACTT-3';
shUBE2R2-1: 5'-CCAATGTCGATGCTTCAGTTA-3';
shScramble: 5'-CCTAAGGTTAAGTCGCCCTCG-3'.

## Establishment of stable cell lines

shRNA plasmids were transfected into HEK 293 T cells with poly-ethylenimine (PEI), according to the manufacturer's instructions. Lentiviral particles produced by HEK 293 T cells were released into the DMEM medium. At 48 and 72 h, the lentiviral particle-containing medium was collected and filtered using a 0.45 μm Syringe Filter Unit. HTR8/SVneo cells were cultured in a 6-well plate (400,000 cells/well) for 24 h to achieve 70–80% confluency at the time of infection by 2 mL lentiviral particle-containing medium. After one day post-infection, 2 days of puromycin selection (2 μg/mL), and knockdown determination, single cells were picked and seeded into 96-well plates to generate monoclonal cell lines.

## Determination of knockdown by quantitative polymerase chain reaction (qPCR)

When infection and puromycin selection of HTR8/SVneo cells were completed or monoclonal cells from 96-well plates were transferred to 6-well plates, total RNA was extracted using the Eastep® Super Total RNA Extraction Kit (Promega). An equivalent of 1 μγ of RNA was reverse transcribed to synthesize cDNA using HiScript® II Q Select RT Super-Mix (Vazyme Biotech). Then, 25 ng of cDNA was used as a template for qPCR analysis with ChamQ Universal SYBR qPCR Master Mix (Vazyme Biotech). The forward and reverse primers for qPCR are listed below:

$qMMUT$-F: 5'-CAGTTGGAAAAAGAAGACGCTGTA-3';
$qMMUT$-R: 5'-ATCTGCCTGTTTCGCACTGA-3';
$qRRM1$-F: 5'-ACTAAGCACCCTGACTATGCTATCC-3';
$qRRM1$-R: 5'-CTTCCATCACATCACTGAACACTTT-3';
$qKIAA1429$-F: 5'-GTTGTGCCACCACCAAGAGG-3';
$qKIAA1429$-R: 5'-AACCCACCACGGGAAGAAAT-3';
$qYWHAZ$-F: 5'-AGCCATTGCTGAACTTGATACA-3';
$qYWHAZ$-R: 5'-AATTTTCCCCTCCTTCTCCTG-3';
$qGAPDH$-F: 5'-TGACAACGAATTTGGCTACA-3';
$qGAPDH$-R:5'-GTGGTCCAGGGGTCTTACTC-3'.

## Reporting summary

Further information on research design is available in the Nature Portfolio Reporting Summary linked to this article.

## Data availability

The processed RNA-seq data, full summary statistics of eQTL and GWAS meta-analysis generated in this study are available at Figshare (https://figshare.com/s/f6de1a56ad7c448c1f4c). The TRF-based TL measurement of placenta samples generated in this study are provided in the Source Data. The raw genotyping and RNA-seq data are

protected and are not available due to data privacy laws. The individual-level genotypes of UKBB samples are available by application to the UKBB (https://www.ukbiobank.ac.uk/register-apply/). The data associated with the curated genome-wide studies which collected from PubMed and literature, are listed at Supplementary Data. The full GWAS summary statistics for TOPMed are available in the database of Genotypes and Phenotypes (dbGaP), under accession code phs001974.v3.p1. The full GWAS summary statistics for SCHS are available at Figshare (https://doi.org/10.6084/m9.figshare.8066999). The full GWAS summary statistics for UKBB data used in this study are available in the https://figshare.com/s/caa99dc0f76d62990195. The TL data of various tissues in GTEx are available in the (https://www.gtexportal.org/home/datasets). The WGS data of GTEx Whole blood samples are available in the dbGaP, under accession code phs000424.v8.p2. Source data are provided with this paper.

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

## Acknowledgements

This study was supported by the National Natural Science Foundation of China (Grant Numbers 32270717, 32070675, 32170762, 82001579) and the Tianjin Committee of Science and Technology (Grant Numbers 19JCJQJC63600 and 20JCYBJC01400), and Tianjin Key Laboratory of Human Development and Reproductive Regulation (2019XHY06 and 2019XHY07).

## Author contributions

D.H., F.W., M.J.L. and Y.C. conceived of the project. Y.Z., J.Z. and D.H. performed the experiments and analyses. W.L., J.C., Y.J., Sha.Z., Y.S., Q.L., X.F., H.Y., X.D., Shi.Z. and X.Y. contributed the data collection and processing. Le.S., Li.S., Z.L., J.Y., J.H., X.M. and K.C. contributed to manuscript polishing and provided analysis suggestions. D.H., M.J.L., Y.Z., F.W., Y.C. and J.Z. wrote the manuscript. All authors read and approved the final submission.

## Competing interests

The authors declare no competing interests.

## Additional information

**Supplementary information** The online version contains
supplementary material available at

Mulin Jun Li, Feng Wang or Dandan Huang.

**Peer review information** *Nature Communications* thanks Veryan Cod-
dand the other, anonymous, reviewer(s) for their contribution to the peer
review of this work. A peer review file is available.

[1]Tianjin Key Lab of Human Development and Reproductive Regulation, Tianjin Central Hospital of Obstetrics and Gynecology, Nankai University,
Tianjin, China. [2]Department of Bioinformatics, The Province and Ministry Co-sponsored Collaborative Innovation Center for Medical Epigenetics, Key
Laboratory of Immune Microenvironment and Disease (Ministry of Education), School of Basic Medical Sciences, Tianjin Medical University, Tianjin, China.
[3]Department of Genetics and Tianjin Key Laboratory of Cellular and Molecular Immunology, School of Basic Medical Sciences, Tianjin Medical University,
Tianjin, China. [4]Department of Clinical Laboratory, Shanghai Children's Hospital, Shanghai Jiaotong University, Shanghai, China. [5]Department of Phar-
macology, Tianjin Key Laboratory of Inflammation Biology, School of Basic Medical Sciences, Tianjin Medical University, Tianjin, China. [6]State Key Laboratory
of Medicinal Chemical Biology, College of Pharmacy, Tianjin Central Hospital of Gynecology Obstetrics/Tianjin Key Laboratory of Human Development and
Reproductive Regulation, Nankai University, Tianjin, China. [7]Key Laboratory of Immune Microenvironment and Disease (Ministry of Education), School of
Basic Medical Sciences, Tianjin Medical University, Tianjin, China. [8]Wuxi School of Medicine, Jiangnan University, Wuxi, China. [9]Department of Pancreatic
Cancer, Key Laboratory of Cancer Prevention and Therapy, National Clinical Research Center for Cancer, Tianjin Medical University Cancer Institute and
Hospital, Tianjin, China. [10]Department of Epidemiology and Biostatistics, Tianjin Key Laboratory of Molecular Cancer Epidemiology, National Clinical
Research Center for Cancer, Tianjin Medical University Cancer Institute and Hospital, Tianjin Medical University, Tianjin, China. [11]Tianjin Medical University
School of Stomatology, Tianjin Medical University, Tianjin, China. [12]Department of Geriatrics, Tianjin Medical University General Hospital; Tianjin Geriatrics
Institute, Tianjin, China. [13]These authors contributed equally: Ying Chang, Yao Zhou, Junrui Zhou. ✉e-mail: mulinli@connect.hku.hk; wangf@tmu.edu.cn;
mikey.huang2011@gmail.com

