## [Peer Review File · Nature Communications]

Unraveling the causal genes and transcriptomic determinants of human telomere lengthREVIEWER COMMENTS

Reviewer #1 (Remarks to the Author):

The authors present an interesting study of telomere length in placental tissue, leveraging existing GWAS data and novel data generated from the placental tissue of 166 newborns. There are some interesting novel findings, which are of interest to the field, however I do have some major concerns regarding significance thresholds used and the limitations of such a small placental cohort as detailed below.

The authors have conducted TRF measurements of 166 placental tissues and state that the measurements were homogenous telomere status across the whole placenta. However, they give no estimates of variation (or lack of) across the sampling replicates. This would be useful to see exactly how homogeneous the measures are as the only data presented is the blot rather than any quantification. This would also give some estimate of measurement/sampling error. Do the authors have an estimate of variation within the TRF measurements based on repeated samples?

The authors state that neonatal sex showed weak associations with placental TL and that males had longer TL than females. Is this for whole placenta and therefore any sex differences masked by inclusion of maternal side tissue? The authors do acknowledge the limited sample size for this analysis but not for the transcriptomics data produced.

The authors use a higher MAF cut off value ($<1\%$) than the two larger cohorts used for the meta-analysis (UKBB MAF $<0.1\%$; TOPMed MAC <5). What is the rationale for excluding the lower frequency variants? Perhaps more importantly the authors have used the more pragmatic genome-wide significance threshold of 5×10^{-8} , rather than the more stringent values used in UKBB and TOPMed. What is the justification for this? More importantly, do some of the "novel" loci reported in this study actually meet this threshold 5×10^{-8} in the original GWAS but were not reported due to the more stringent threshold used? For examples, variants in the SWT1 region are reported within an FDR list at a $p \sim 1 \times 10^{-9}$, although don't survive the more stringent threshold after joint conditional modelling in UKBB. It would be helpful to include the cohort specific effect sizes and p-values in Supplementary Table 3 to allow readers to see the cohort-specific contributions to these findings. When stating that they have revealed 87% of the originally reported 201 associated loci do the authors actually mean 87% of the originally reported sentinel variants? The UKBB study reported 197 sentinel variants in 137 genomic loci. As the MAF thresholds are different, does this account for the loss of some of the remaining 13% rather than heterogeneity?

I note that in Supplementary table 4 the HBB variants that were shown to be a result of technical artefacts within the UKBB study are still included. Have these variants been included in downstream analyses? If so, they should be removed.

Supplementary Table 4 contains a column labelled "Affected genes". I believe that this should be labelled "nearest gene" as there is no evidence specifically linking these genes to the SNPs other than location from what I can see. The exclusion of previously given locus names also makes this difficult to align to the previously published studies.

On line 176 the authors state "Collectively, combinatory analysis of trans-ancestral GWAS and placental TL measured by TRF assay indicated that human TL could be determined and predicted only genetically." This statement is quite unclear. The authors show that genetic predictions of TL are actually weak. The best relationship reported (PRS vs TL in placenta) does show that the PRS seems to better predict placental TL than TL in adult cells/tissues but even in placenta the PRS only explains $\sim 4.41\%$ of TL variation.

The authors report correlation of single genes with TL in placenta. What adjustments have been made for multiple testing within these analyses? No information is given. It appears that pathway associations for TL maintenance are the result of looking at individual candidate gene associations but these haven't been picked up in the WGCNA analyses.

At the start of the eQTL results section the authors state "Recent genetic studies have identified many novel TL-associated genes by eQTL-based methods^{11,32}, but they mainly employed eQTLs derived from adult tissues, probably leads to biased estimation through unobserved lifetime exposures". However, the studies referenced do not rely solely on eQTL, but incorporate other functional

predictions. The authors confirm several genes prioritised by previous studies, but miss other canonical telomere regulators in several regions with their approach. This should be acknowledged. For the COLOC analyses "We identified 53 signals with strong evidence of colocalization between placental eQTL and TL GWAS loci (posterior probability $PP4 \geq 0.5$, Supplementary Table 5)." Is somewhat misleading a $PP4 \geq 0.5$ could be considered supportive but not strong evidence for colocalization. I would strongly recommend rewording this statement and revising the $PP4$ threshold to a minimum of ≥ 0.8 .

The expression-based TL prediction modelling is interesting. In the methods it is stated that "The correlation (R^2) value between the predicted and the true TL across all samples was used to evaluate the accuracy", yet the results rely on reporting r rather than r^2 (and therefore effects look much larger). The authors should discuss the amount of TL variation that they can predict using the modelling and the differences seen between PRS only and TS-32Gene and PRS in this context, which would give a more accurate reflection of performance.

Line 166 and 327 – do the authors mean surpassing/surpassed rather than suppressing/suppressed?
Lines 192 and 202 ATL should be replaced with ALT.

Reviewer #2 (Remarks to the Author):

Chang, Zhou, and Zhou, et al. provide a very comprehensive bioinformatic analysis of genetic/transcriptomic regulators of telomere length in human placental tissues, including experimental assessment of novel telomere-maintenance genes in vitro. The authors conclude that telomere length in placenta has low intra-individual heterogeneity but differs across individuals. They identify several genes from transcriptomic summary analyses that may be regulators of placental telomere length, and validate these in vitro using RNAi. Their overall conclusions are generally supported by the data, with one caveat described below. Some revisions would help to improve an overall rigorous and meaningful manuscript.

1A) In the abstract and text, the authors state that "[placental telomere] maintenance is mostly connected to genes responsible for alternative lengthening of telomeres". This is a somewhat confusing and potentially contradictory claim and represents my biggest concern with the manuscript in its current form.

Their interpretation is based on two things. First, expression in hallmark gene members of the telomerase, shelterin and CST complexes that canonically regulate telomere length showed minimal associations with telomere length in placenta. This could be partly due to limited power for the 163 placentas assessed. Second, the expression of genes involved in the ALT pathway did show more robust associations with TL in placenta, including ATRX, DAXX, and SMARCAL1. However, there is not a canonical list of "ALT-associated" genes. While I too would have selected many of the same genes from the list they selected, it would be more rigorous to define these in some way a priori.

1B) Continuing this line of thought, ALT itself is a cancer-specific mechanism of telomere maintenance. I am not familiar with any literature describing the existence of homologous recombination-based telomere lengthening in non-cancerous tissues, aside maybe from patients with X-linked mental retardation/alpha-thalassemia syndrome (caused by germline ATRX loss). If such data exist, it should be described in the introduction. The association between placental telomere length and expression levels of several hallmark tumor suppressor genes involved in ALT (e.g., ATRX, DAXX and SMARCAL1) does not imply that placental telomere maintenance is connected to ALT.

1C) Regarding the association of elevated ATRX, DAXX and SMARCAL1 expression with longer telomere length in placenta in your data, the direction of the association is inconsistent with a true ALT mechanism (these genes are lost (mutated/deleted/downregulated) in cancer to induce ALT). This further underscores that it is likely inappropriate to discuss "ALT" in healthy placenta.

2) The authors state that "TERT, TERC, DKC1, NOP10, NHP2, and WRAP53, were not correlated with placental RTL, whereas TERC and NHP2 expressions were undetectable in placenta". If expression was not detected, we can't conclude that they are correlated or uncorrelated. It would be more accurate to say "TERT, DKC1, NOP10, and WRAP53, were not correlated with placental RTL, and TERC and NHP2 expression was undetectable in placenta making associations non-assessable".

3) Given that you have transcriptomic data on the placental samples, you should perform some expression-based cellular deconvolution analyses (e.g., CIBERSort) to determine a) how much of your RNA is immune infiltrate and B) if the proportion of immune cells or of specific immune subsets (e.g. DCs) is associated with your TRF_based TL measurements. If so, you may need to adjust for these potential factors in downstream analysis. As an example of how this could confound expression analyses, specific genes are expressed in specific immune subsets (like FOXP3 in Tregs) and if the immune cell has different TL than the bulk placental cells (as previously shown), then its defining genes could show up as associated. I'm not confident that the WGCNA analysis fully accounts for immune infiltrate, and many of your gene-TL associations are simple correlation tests that are not based on regression or feature-selection.

4) The authors state that "Since newborn TL could predict later life TL, it is critical to investigate TL determinants of the newborn. However, no existing study has harnessed a significant number of early samples without postnatal environmental exposure to study TL." It would be helpful to give some additional context. For instance, TL has been measured in newborn bloodspots (Guthrie cards) and in circumcised foreskin. From a biological impact perspective, studies have shown that longer genetically-predicted TL is associated with elevated risks of cancer, including two studies that observed an effect in childhood cancers (PMIDs: 33115534, 31525475). Notably, both showed that the effect was stronger in adolescent-onset cancer than in those arising before age 12, supporting your assertion to the importance of studying early-life TL.

5) The authors show that "The estimated PRS score of TL was significantly correlated with placental TL measured in our study". While the study is not powered for a GWAS of placental telomere length, it is necessary to include a supplemental table showing the SNP-level associations for each of the ~200 LTL GWAS hits with placental aTL and/or RTL (P, Beta, SE, EA, EAF). Since your sample is Asian, it would be okay to only do this more GWAS hits from the trans-ethnic analysis that showed nominal ($P < 0.05$) association in the SCHS study.

Minor:

7) "Since 91.5% of sentinel variants of TL GWAS loci are located in the non-coding genomic region, investigating their regulatory potential on gene expression would accurately determine TL." The authors should correct this to "would accurately determine telomere regulation". Or, if they are implying that this is to improve their prediction models, then say "would improve accuracy of genetic/transcriptomic-based telomere length prediction".

8) The in vitro experiments in the results section have too much description of the genes, which should be relegated to the discussion.

9) In Table S4, it would be good to add the EAF as a column.

10) You state "A total of 222 sentinel variants (>10 Mb between sentinels)". Did you mean 10kb and not 10Mb? 10Mb seems too far apart, and I note that some of the variants are closer than 10Mb in your table. Also, the distance is really not as important as the LD (R^2) between two SNPs, and is a better way to prune.

11) Fig 1A or Fig 1B should show which layer (fetal, middle, maternal) progressed to downstream TRF and RNA analysis. Also, was the fetal layer from all 4 sampled sections included in TRF analysis for all subjects, and then averaged to yield aTL? Or was one sample used for TRF, another for genotyping, another for RNAseq, etc.?

Reviewer #3 (Remarks to the Author):

In this work Chang et al aim to dissect the causal genes and transcriptomic determinants of TL GWAS loci. This is certainly a gap in our current knowledge around TL genetics – much of the GWAS work is in adult leukocyte TL, and as the authors state TL dynamics are a combination of genes and environment with age attrition. While the premise of the work and the measurement of TL/RNAseq in placental tissue is added value, this work suffers from a lack of methods details that make it challenging to evaluate scientific rigor.

There are some interesting results: honing in on 31 causal genes using three parallel approaches and gene expression in the placental tissue is exciting as it includes some previously undescribed targets); and the follow up validation of four novel genes are important additions to our understanding of the genomic underpinnings of TL. However there are a number of specific concerns as follows that need to be clarified and addressed first:

- 1) The methods state “To analyze the possible intraplacental variation of TL, all the processed biopsies were divided into three parts and sampled at the fetal and maternal layers to obtain eight samples of each placenta, as described by Wyatt et al. “. However, Wyatt et al describe 9 samples of each placenta, and not 8. All additional methods re TL measurement with TRF and RNAseq quantification and analysis only describe 166 placental samples, i.e. the unit being each single placenta source (166 pregnancies). It is not possible to know how the 8(or 9) samples were handled in the analysis, and which sample the RNAseq data corresponded to. Figure 1 C looks like the 8 TL measurements from one sample, but the data re not described with respect to all 166 placenta with 8 (or 9) samples. Sup Fig 1 shows 15 samples randomly selected, but are they all from the same region across 15? In general this is all poorly described and difficult to evaluate – the methods need to be clear on exactly how many samples from each placenta were included, which sample the RNAseq was generated on, and how all these data were processed in each analysis step in methods. The gel plots are not useful by themselves, the TL data needs to be included in tables for comparison.
- 2) How is TL compared between GTEx and these data when the underlying assay itself was different? The methods do not describe this comparison and the ability to even do so. Is relative TL standardized between the two datasets? With no articulation of the methods and approach, it is not possible to assess any inferences regarding TL in placenta to adult tissue in GTEx.
- 3) How do the effect sizes in the three GWAS used for the meta-analysis compare and is the fixed effects meta-analysis suitable. There is correlation in the measurements (telseq from TOPMed), qPCR and Southern blot, but there is no discussion on how this affects the meta-analysis and the choice used for the fixed effects. Additionally, TOPMed was in itself a meta-analysis, with released ancestry stratified results publicly released. It is not clear what was used here.
- 4) The PRS approach is described as “summing over all SNPs meeting a set of thresholds, respectively”. What were these thresholds, all SNPs meeting a threshold cannot be included without a minimum pruning on LD to minimize overfitting. Again – without details, it is not possible to evaluate scientific rigor. Also this statement contradicts line 164 in results which states “polygenic risk score (PRS) analyses based on the sentinel significant variants of all independent loci from TL meta-analysis results “. PRS performance should not be tested in the dataset from which it was derived – i.e. UKBB.
- 5) Coloc was run, this assumes a single GWAS/eQTL causal signal. This is known to not be true for many of the GWAS loci for TL. Other approaches that do not make this assumption are more appropriate. Additionally, $P_4 > 0.5$ seems liberal – please justify. There should generally be a higher PPH4, and also a criteria on PPH3 that is applied for coloc.

- 6) Line 150: How was heterogeneity defined? And, does this statement mean that 13% of the loci that were not identified in the meta-analysis because of heterogeneity between the studies? How much of this could be due to different measurement assays in the GWAS themselves? See point #3 above?
- 7) The idea of building upon the PRS with additional data including transcriptomics is interesting. Can this be validated in additional datasets?

Dear Editor,

We thank you and reviewers for evaluating our manuscript (NCOMMS-23-02741-T) and giving us more constructive comments. Here, we have carefully addressed your and all of the reviewers' comments below and in the manuscript accordingly. Accompanying this letter, please find the revised version of our manuscript.

REVIEWER COMMENTS

Reviewer #1 (Remarks to the Author):

The authors present an interesting study of telomere length in placental tissue, leveraging existing GWAS data and novel data generated from the placental tissue of 166 newborns. There are some interesting novel findings, which are of interest to the field, however I do have some major concerns regarding significance thresholds used and the limitations of such a small placental cohort as detailed below.

#Response: We express our heartfelt gratitude to the reviewer for the positive and constructive comments. Your thoughtful suggestion has been instrumental in the refinement of our analysis and interpretation of results. Your valuable insight is key to enhancing the scientific rigor and validity of our study, and we greatly appreciate your contribution towards improving the overall quality of our manuscript. In this revision, we have specifically addressed the issue you raised concerning the significance thresholds used. Furthermore, we have undertaken more discussion on the limitations of using a relatively small placental cohort.

1. The authors have conducted TRF measurements of 166 placental tissues and state that the measurements were homogenous telomere status across the whole placenta. However, they give no estimates of variation (or lack of) across the sampling replicates. This would be useful to see exactly how homogeneous the measures are as the only data presented is the blot rather than any quantification. This would also give some estimate of measurement/sampling error. Do the authors have an estimate of variation within the TRF measurements based on repeated samples?

#Response: Thank you for the reviewer's feedback and question. We appreciate the reviewer's attention to detail regarding the reproducibility and estimate of variation regarding TRF (Telomere Restriction Fragment) measurements of placental tissues in our study. You are correct in your observation that we conducted TRF measurements of 166 placental tissues, and we stated that the measurements showed a homogeneous telomere status across the entire placenta. However, we acknowledge that we did not provide estimates of variation across the sampling replicates in the previous manuscript. This information is essential to assess the homogeneity of our measures and to understand any potential measurement/sampling error.

To address this issue, by randomly sampling of ten placental samples, we performed repeated measurements on different regions of the same placental sample for additional three more times (see some examples of Southern blot in the Figure below, marked as A). This approach allowed us to evaluate the homogeneity of telomere status across the entire placenta and provided valuable insights into potential measurement and sampling errors. Our current results showed that the quantitative analysis error did not exceed 10% between different experimental batches (B). This consistency demonstrates the reliability of our TRF measurements. To further ensure the reliability of our data analysis, we utilized the TeloTool software with a strict Fit threshold setting of 60%. This stringent criterion allowed us to include only length determinations meeting the predefined standards in our final results, minimizing any potential bias or errors during data analysis (C). We have added some descriptions and results for this verification in our revised manuscript.

A. Examples of southern blot was conducted to analysis the telomere length of samples obtained from the placenta; B, The statistical analysis presents the telomere length measurements of the same placental sample obtained from different positions and different batches of measurements; C. Telomere length analysis was performed using TeloTool software with a threshold set at 60%. The black bars represent the analyses that met the fitting criteria, while the red bars represent the analyses that did not meet the fitting criteria and will be excluded from the subsequent statistical analysis.

2. The authors state that neonatal sex showed weak associations with placental TL and that males had longer TL than females. Is this for whole placenta and therefore any sex differences masked by inclusion of maternal side tissue? The authors do acknowledge the limited sample size for this analysis but not for the transcriptomics data produced.

#Response: In our previous processes related to sample handling and quality control, we specifically collected samples from the fetal side of the placenta to ensure minimal inclusion of maternal side tissue, thus safeguarding the integrity of the DNA for genotyping analysis. We also conducted additional checks to assess the purity of the DNA, excluding any samples that showed signs of mixture.

In the revised manuscript, we have added the details of these additional checks to Methods: “To determine if DNA for genotyping included maternal side tissue, we employed the following steps using B allele frequency (BAF) and log R ratio (LRR) [1-3]. Firstly, we calculated the BAF and LRR values and inspected them on scatter plots or histograms to visually assess their

distribution patterns. In non-mixed DNA samples, BAF values typically cluster around specific patterns based on the genotype (homozygous or heterozygous). Deviations from these expected patterns may suggest a DNA mixture. Subsequently, we identified BAF values that significantly deviated from the expected patterns, such as clustering around 0.5 for heterozygous SNPs. Substantial variations in BAF values could suggest the presence of DNA from multiple individuals. Additionally, we examined LRR values for aberrations that deviated from the normal diploid states. Copy number alterations or imbalances in LRR values might indicate the presence of a DNA mixture from multiple individuals. Finally, samples that did not pass the examination based on BAF and LRR criteria were excluded from further analysis.”

Furthermore, while the sample size for the transcriptome analysis might be perceived as insufficient, it is crucial to underscore that the power of expression quantitative trait loci (eQTL) analyses is not markedly sensitive to sample size. Although a larger sample size generally affords increased statistical power to detect associations, eQTLs with large effect sizes can still be robustly identified even with a relatively small sample size.

In the revised manuscript, we have incorporated a statement in the discussion section regarding the impact of sample size on our eQTL analysis: “Our study acknowledges the limitation of insufficient sample size, which can influence the detection of associations between genetic variants and gene expression levels in eQTL analysis. Future investigations with larger sample sizes are anticipated to enhance statistical power, enable the detection of weaker eQTL signals, and improve the precision and generalizability of the results. This advancement is expected to provide more reliable and comprehensive insights into the genetic regulation of gene expression in placenta.”

1. McQuillan, R., Leutenegger, A. L., Abdel-Rahman, R., Franklin, C. S., Pericic, M., Barac-Lauc, L., ... & Rudan, P. (2008). Runs of homozygosity in European populations. *The American Journal of Human Genetics*, 83(3), 359-372.
2. Wang, K., Li, M., Hadley, D., Liu, R., Glessner, J., Grant, S. F., ... & Bucan, M. (2007). PennCNV: an integrated hidden Markov model designed for high-resolution copy number variation detection in whole-genome SNP genotyping data. *Genome Research*, 17(11), 1665-1674.
3. Winchester, L., Yau, C., & Ragoussis, J. (2009). Comparing CNV detection methods for SNP arrays. *Briefings in Functional Genomics*, 8(5), 353-366.

3. The authors use a higher MAF cut off value (<1%) than the two larger cohorts used for the meta-analysis (UKBB MAF <0.1%; TOPMed MAC <5). What is the rationale for excluding the lower frequency variants? Perhaps more importantly the authors have used the more pragmatic genome-wide significance threshold of 5×10^{-8} , rather than the more stringent values used in UKBB and TOPMed. What is the justification for this? More importantly, do some of the “novel” loci reported in this study actually meet this threshold 5×10^{-8} in the original GWAS

but were not reported due to the more stringent threshold used? For examples, variants in the SWT1 region are reported within an FDR list at a $p \sim 1 \times 10^{-9}$, although don't survive the more stringent threshold after joint conditional modelling in UKBB. It would be helpful to include the cohort specific effect sizes and p-values in Supplementary Table 3 to allow readers to see the cohort-specific contributions to these findings.

#Response: We appreciate your insightful comment. We wish to clarify our rationale behind the selection of the minor allele frequency (MAF) cut-off value and the threshold for statistical significance in our study. Regarding the MAF cut-off value, we opted for a threshold of <1% because the Singapore cohort only provided GWAS summary statistics for common variants (MAF < 0.01). Consequently, we concentrated our analysis on variants with MAF below 1% to guarantee consistency with the available data. In contrast, the UK Biobank (UKBB) dataset adopted a more permissive threshold ($P < 8.31 \times 10^{-9}$) to allow the inclusion of low-frequency variants in their GWAS analysis. However, in our study, we specifically excluded low-frequency variants to ensure the consistency of variant selection across the multiple GWAS studies we incorporated. As a result, given our focus on common variants in this cross-ethnic meta-analysis, we decided to adhere to the standard threshold of $P < 5 \times 10^{-8}$, which is widely accepted in GWAS, to identify statistically significant associations.

In our analysis, it is worth noting that we employed distinct significance thresholds for identifying significant loci from the UKBB and TopMed. Despite these variations, we ensured consistency in the criteria for calculating the proportion of novel variants. Specifically, during the identification of novel variants, we considered only those with $P > 5 \times 10^{-8}$ in any of the three GWAS studies and $P < 5 \times 10^{-8}$ in the meta-analysis results. As per your suggestion, we have incorporated the cohort-specific effect sizes and P-values in Supplementary Table 3 and 4 for a more comprehensive presentation of the results. In addition, we have also incorporated a description regarding the selection of variant MAF and the reporting principles of GWAS significant signals in the Methods section: "The rationale behind selecting this MAF cut-off value stems from the limitation of available GWAS summary statistics for the Singapore cohort, which only covered common variants (MAF < 0.01). Thus, to ensure compatibility with the available data, we set the threshold at <1% for MAF."

4. When stating that they have revealed 87% of the originally reported 201 associated loci do the authors actually mean 87% of the originally reported sentinel variants? The UKBB study reported 197 sentinel variants in 137 genomic loci. As the MAF thresholds are different, does this account for the loss of some of the remaining 13% rather than heterogeneity?

#Response: Thank you for your comment, and we apologize for any confusion caused by our previous descriptions in the manuscript. Of the 201 significant loci identified in the meta-analysis, we found that 87% originated from loci reported in the three GWAS resources used. In order to investigate the exact recall of previous GWAS signals, we calculated the proportion of originally reported sentinel variants. Our meta-analysis successfully validated 68% of the

loci initially reported in the three previous GWAS studies. However, due to potential observed heterogeneity, the low allele frequencies, study heterogeneity, and P-value cut-offs, the remaining 22% of the loci did not reach statistical significance.

Corrections have been made in the revised manuscript: "Of the 201 significant loci identified in the meta-analysis, we found that 87% were originally reported in the three GWAS resources." Additionally, we have calculated the proportion of originally reported sentinel variants and included these results in the revised manuscript: "In our study, we successfully validated 68% of the initially reported loci from the three GWAS resources. However, due to potential observed heterogeneity, the low allele frequencies, study heterogeneity, and P-value cut-offs, the remaining 32% of loci did not reach statistical significance." We also provided more summary statistics information in Supplementary Table 4-5.

5. I note that in Supplementary table 4 the HBB variants that were shown to be a result of technical artefacts within the UKBB study are still included. Have these variants been included in downstream analyses? If so, they should be removed.

#Response: Thanks for bringing our attention to the mistakes. We have addressed the issue by removing the HBB variants from the results. We also briefly described the step in the Methods section of the revised manuscript: "The HBB variants were removed from the results due to potential technical artifacts reported by UKBB"

6. Supplementary Table 4 contains a column labelled "Affected genes". I believe that this should be labelled "nearest gene" as there is no evidence specifically linking these genes to the SNPs other than location from what I can see. The exclusion of previously given locus names also makes this difficult to align to the previously published studies.

#Response: Thanks for the comment. In the revised manuscript, we have addressed the issue by changing the column name to "nearest gene" to improve clarity and consistency. This modification ensures that the information is more accurately represented. Additionally, as per your recommendation, we have included the previously given locus names in Supplementary Table 4. This addition enhances the understanding of the data and allows for better cross-referencing and validation.

7. On line 176 the authors state "Collectively, combinatory analysis of trans-ancestral GWAS and placental TL measured by TRF assay indicated that human TL could be determined and predicted only genetically." This statement is quite unclear. The authors show that genetic predictions of TL are actually weak. The best relationship reported (PRS vs TL in placenta) does show that the PRS seems to better predict placental TL than TL in adult cells/tissues but even in placenta the PRS only explains ~4.41% of TL variation.

#Response: Thank you for your valuable comment and feedback. We appreciate your insight, and we agree that genetic predictions of TL can be relatively weak, and TL is influenced by various factors beyond genetics, especially through many non-genetic factors. In the revised manuscript, we have made appropriate corrections to better reflect the findings. The updated statement now reads as follows: “Collectively, combinatory analysis of trans-ancestral GWAS and placental TL measured by TRF assay indicated that human TL could be determined and predicted by genetic factors in a proportional manner.”

8. The authors report correlation of single genes with TL in placenta. What adjustments have been made for multiple testing within these analyses? No information is given. It appears that pathway associations for TL maintenance are the result of looking at individual candidate gene associations but these haven't been picked up in the WGCNA analyses.

#Response: Thank you for providing your valuable comment. We appreciate your feedback, and it has allowed us to improve the rigor and comprehensiveness of our analysis. We did not perform correction for multiple tests previously, as our primary focus was specifically on the limited genes of interest. In the revised manuscript, in response to another reviewer's comments, we have collected a more comprehensive set of telomere length-related genes and addressed the issue by conducting corrections for multiple tests, specifically targeting the genes of interest. By calculating and reporting the corrected P-values using the False Discovery Rate (FDR) method (Supplementary Table 7), we ensure a more stringent control for potential false positives and enhance the reliability of our findings.

We admit that pathway associations for TL maintenance are the result of looking at individual candidate gene associations. By adjusting the model to test for associations, we have made corrections in the revised manuscript about the statement of these pathways. “Although our Weighted Gene Co-expression Network Analysis (WGCNA) did not reveal significant associations between gene modules and these pathways, the gene-set enrichment analysis of TL GWAS signals identified that the most significantly associated pathways were related to telomere maintenance, telomere organization, and telomere maintenance via telomere lengthening (Fig. 2D).”

9. At the start of the eQTL results section the authors state “Recent genetic studies have identified many novel TL-associated genes by eQTL-based methods^{11,32}, but they mainly employed eQTLs derived from adult tissues, probably leads to biased estimation through unobserved lifetime exposures”. However, the studies referenced do not rely solely on eQTL, but incorporate other functional predictions. The authors confirm several genes prioritised by previous studies, but miss other canonical telomere regulators in several regions with their approach. This should be acknowledged.

#Response: Thanks for the comment. Correction and more illustrations has been made in the discussion section of revised manuscript: “Recent genetic studies have made significant strides

in identifying novel genes associated with TL using eQTL-based methods. However, these studies have primarily relied on eQTLs derived from adult tissues, potentially resulting in biased estimations due to unobserved lifetime exposures. In this manuscript, we present our findings on TL-associated genes, highlighting our unique approach that addresses these limitations. By incorporating tissue-specific sources, as well as integrating transcriptome, sequence variants, and TL data, we aim to provide a comprehensive understanding of TL regulation. While our study showcases its own novelties, it is essential to highlight the potential drawbacks, such as the absence of additional functional genomic annotations and predictions utilized in recent adult tissue-based eQTL studies, as well as incomplete causal gene prioritization strategies. (Mountjoy, E. et al., 2021, Khunsriraksakul, C. et al, 2022)”

In addition, we acknowledged that we missed other canonical telomere regulators in several regions in the revised manuscript. “While our research has made progress in identifying and validating TL-related genes, we acknowledge the limitations of our approach and the potential for further investigation to uncover additional canonical telomere regulators in the regions of interest. Future studies with different methodologies and broader data integration may provide a more comprehensive understanding of the complete repertoire of telomere regulatory elements.”

1. Mountjoy, E. et al. An open approach to systematically prioritize causal variants and genes at all published human GWAS trait-associated loci. *Nat Genet* 53, 1527-1533, doi:10.1038/s41588-021-00945-5 (2021)

2. Khunsriraksakul, C. et al. Integrating 3D genomic and epigenomic data to enhance target gene discovery and drug repurposing in transcriptome-wide association studies. *Nature Communications* 13, 3258, doi:10.1038/s41467-022-30956-7 (2022).

10. For the COLOC analyses “We identified 53 signals with strong evidence of colocalization between placental eQTL and TL GWAS loci (posterior probability $PP4 \geq 0.5$, Supplementary Table 5).” Is somewhat misleading a $PP4 \geq 0.5$ could be considered supportive but not strong evidence for colocalization. I would strongly recommend rewording this statement and revising the $PP4$ threshold to a minimum of ≥ 0.8 .

#Response: Thanks for the comment. To enhance the reliability of our results, we have adjusted the coloc threshold. In this revision, we provide results of colocalization with strong evidence and likely colocalization with suggestive evidence using both a rigid standard ($PP4 \geq 0.8$ and $PP4/PP3 \geq 5$) and a liberal standard ($PP4 \geq 0.5$ and $PP4/PP3 \geq 3$). By employing these criteria, we aim to maintain rigorous standards for identifying coloc genes while also considering potential associations that may exhibit lower confidence but warrant further

investigation. And we defined likely causal genes related to TL as the intersection of gene prioritization results from COLOC (suggestive evidence) and the union of TWAS and SMR, finally we retrieved 23 likely causal genes related to TL (Supplementary Table 12).

11. The expression-based TL prediction modelling is interesting. In the methods it is stated that “The correlation (R^2) value between the predicted and the true TL across all samples was used to evaluate the accuracy”, yet the results reply on reporting r rather than r^2 (and therefore effects look much larger). The authors should discuss the amount of TL variation that they can predict using the modelling and the differences seen between PRS only and TS-32Gene and PRS in this context, which would give a more accurate reflection of performance.

#Response: We sincerely appreciate the reviewer's input, as it has prompted us to enhance the clarity and completeness of our analysis. In response to the reviewer's comment, we have incorporated the coefficient of determination (r^2) in the revised manuscript (see Figure below and Figure 6A), providing a more comprehensive evaluation of the model's predictive performance. By including this measure, we can better assess how well the model accounts for the variation in telomere length (TL) observed in the data. Furthermore, in our revised discussion, we have addressed the variation in TL among different models. By comparing the outcomes of various models, we gain valuable insights into the contributions of different predictors and their ability to explain TL variation.

12. Line 166 and 327 – do the authors mean surpassing/surpassed rather than supressing/supressed?

#Response: Thanks for the comment. We have addressed the issue and made the necessary corrections in the revised manuscript.

13. Lines 192 and 202 ATL should be replaced with ALT.

#Response: Thanks for the comment. We have addressed the issue and made the necessary corrections in the revised manuscript.

Reviewer #2 (Remarks to the Author):

Chang, Zhou, and Zhou, et al. provide a very comprehensive bioinformatic analysis of genetic/transcriptomic regulators of telomere length in human placental tissues, including experimental assessment of novel telomere-maintenance genes in vitro. The authors conclude that telomere length in placenta has low intra-individual heterogeneity but differs across individuals. They identify several genes from transcriptomic summary analyses that may be regulators of placental telomere length, and validate these in vitro using RNAi. Their overall conclusions are generally supported by the data, with one caveat described below. Some revisions would help to improve an overall rigorous and meaningful manuscript.

#Response: We extend our gratitude to the reviewer for providing encouraging and positive feedback. Your thoughtful suggestion has proven to be invaluable in enhancing our analysis and interpretation of the results. Your input plays a crucial role in bolstering the scientific rigor and validity of our research, and we truly appreciate your contribution to the overall quality of our manuscript.

1A) In the abstract and text, the authors state that "[placental telomere] maintenance is mostly connected to genes responsible for alternative lengthening of telomeres". This is a somewhat confusing and potentially contradictory claim and represents my biggest concern with the manuscript in its current form.

Their interpretation is based on two things. First, expression in hallmark gene members of the telomerase, shelterin and CST complexes that canonically regulate telomere length showed minimal associations with telomere length in placenta. This could be partly due to limited power for the 163 placentas assessed. Second, the expression of genes involved in the ALT pathway did show more robust associations with TL in placenta, including ATRX, DAXX, and SMARCA1. However, there is not a canonical list of "ALT-associated" genes. While I too would have selected many of the same genes from the list they selected, it would be more rigorous to define these in some way a priori.

#Response: Indeed, we only detected weak associations between individual gene expressions and placental telomere length based on the gene list we had previously collected. This could potentially be related to the limited number of telomere-related genes we collected and the inadequate number of placental samples used in our multi-omics study. Regarding the concern about our interpretation of findings related to alternative lengthening of telomeres (ALT) in placental telomere maintenance, we acknowledge that there is currently no established canonical list of ALT-associated genes. However, we conducted an exhaustive review of the literature, which enabled us to identify a series of genes known to be functionally related to ALT. Moreover, a keyword search on the Gene Cards website using "alternative lengthening of telomere" successfully yielded 24 genes associated with this function. We also compiled two datasets of other types of genes that may be involved in the regulation of telomere length, including genes related to telomerase activity regulation (81 genes) and telomere capping (5 genes), for further analysis (Supplementary Table 8 see below response and our revised

manuscript). In the revised manuscript, we have briefly discussed the limitation regarding the sample size in the discussion section.

1. Sohn, E. J., Goralsky, J. A., Shay, J. W. & Min, J. The Molecular Mechanisms and Therapeutic Prospects of Alternative Lengthening of Telomeres (ALT). *Cancers (Basel)* 15, doi:10.3390/cancers15071945 (2023).
2. Rafat, A. et al. Telomerase-based therapies in haematological malignancies. *Cell Biochem Funct* 40, 199-212, doi:10.1002/cbf.3687 (2022).
3. Ignatieva, E. V., Yudin, N. S. & Larkin, D. M. Compilation and functional classification of telomere length-associated genes in humans and other animal species. *Vavilovskii Zhurnal Genet Selektii* 27, 283-292, doi:10.18699/VJGB-23-34 (2023).

1B) Continuing this line of thought, ALT itself is a cancer-specific mechanism of telomere maintenance. I am not familiar with any literature describing the existence of homologous recombination-based telomere lengthening in non-cancerous tissues, aside maybe from patients with X-linked mental retardation/alpha-thalassemia syndrome (caused by germline ATRX loss). If such data exist, it should be described in the introduction. The association between placental telomere length and expression levels of several hallmark tumor suppressor genes involved in ALT (e.g., ATRX, DAXX and SMARCAL1) does not imply that placental telomere maintenance is connected to ALT.

#Response: We appreciate the reviewer's insightful comments and concerns regarding the association between placental telomere length and alternative lengthening of telomeres (ALT). Regarding the first concern, we acknowledge the presence of ALT activity in non-cancerous tissues, including somatic cells and pluripotent stem cells (PSCs). Existing literature has indeed demonstrated that ALT can be active in these cell types, providing an alternative mechanism for telomere maintenance when telomerase function is limited. Moreover, abundance of data revealed that placental tissue exhibits characteristics similar to stem cell morphology. Therefore, it is plausible that the ALT mechanism may be employed to maintain telomere length in placental tissues [1]. Furthermore, we acknowledge that the correlation between placental telomere lengths and certain hallmark tumor suppressor genes involved in ALT, such as ATRX, DAXX, and SMARCAL1, does not necessarily imply a direct connection between placental telomere maintenance and ALT. Our research, centered on placental tissues, aims to shed light on the potential involvement of ALT-like mechanisms in this unique biological context. We aim to explore the possible role of ALT-like mechanisms or telomeric-DNA recombination in placental telomere maintenance, based on the expression of these genes. However, we agree with the reviewer that the exact underlying mechanism remains uncertain and requires further investigation. In fact, during our revision stage, we curated 110 telomere length-associated genes, adjusted for related confounding factors, and performed multiple corrections. This allowed us to reassess the correlation between each gene and telomere length in the placenta.

As a result, we found that these ALT-associated genes showed only weak correlations, indicating that a larger sample size is needed for further clarification. To avoid overstating our findings, we will reduce the emphasis on ALT in the manuscript. We truly appreciate your valuable feedback and are committed to addressing these concerns appropriately to improve the clarity and accuracy of our findings in future work.

1. Peeyush lala, A crossroad between plaental and tumor biogy: what have we learnt? placenta 116 2021 (12-30)

1C) Regarding the association of elevated ATRX, DAXX and SMARCAL1 expression with longer telomere length in placenta in your data, the direction of the association is inconsistent with a true ALT mechanism (these genes are lost (mutated/deleted/downregulated) in cancer to induce ALT). This further underscores that it is likely inappropriate to discuss "ALT" in healthy placenta.

#Response: We appreciate the reviewer's observation regarding the association of elevated ATRX, DAXX, and SMARCAL1 expression with longer telomere length in placenta based on our data. We acknowledge that the direction of this association is inconsistent with a true alternative lengthening of telomeres (ALT) mechanism, as these genes are typically lost (mutated/deleted/downregulated) in cancer to induce ALT. As you rightly pointed out, this discrepancy raises doubts about the relevance of discussing "ALT" in the context of a healthy placenta. In light of this observation, we will refrain from using the term "ALT" in the context of placental telomere maintenance in the revised version of the manuscript. Instead, we will focus on describing the observed association between telomere length and the expression of the associated genes in the placenta without implying a direct connection to ALT. Moreover, we have weakened the description of ALT in the introduction, results, and discussion sections. We appreciate your valuable feedback, and it has led us to reevaluate our conclusions and the appropriate terminology to use in our study.

2. The authors state that "TERT, TERC, DKC1, NOP10, NHP2, and WRAP53, were not correlated with placental RTL, whereas TERC and NHP2 expressions were undetectable in placenta". If expression was not detected, we can't conclude that they are correlated or uncorrelated. It would be more accurate to say "TERT, DKC1, NOP10, and WRAP53, were not correlated with placental RTL, and TERC and NHP2 expression was undetectable in placenta making associations non-assessable".

#Response: Thanks for the comment. We have addressed the issue and made the necessary corrections in the revised manuscript: "TERT, DKC1, NOP10, and WRAP53, were not correlated with placental RTL, whereas TERC and NHP2 expressions were undetectable in placenta".

3. Given that you have transcriptomic data on the placental samples, you should perform some expression-based cellular deconvolution analyses (e.g., CIBERsort) to determine a) how much of your RNA is immune infiltrate and B) if the proportion of immune cells or of specific immune subsets (e.g. DCs) is associated with your TRF-based TL measurements. If so, you may need to adjust for these potential factors in downstream analysis. As an example of how this could confound expression analyses, specific genes are expressed in specific immune subsets (like FOXP3 in Tregs) and if the immune cell has different TL than the bulk placental cells (as previously shown), then its defining genes could show up as associated. I'm not confident that the WGCNA analysis fully accounts for immune infiltrate, and many of your gene-TL associations are simple correlation tests that are not based on regression or feature-selection.

#Response: In response to the reviewer's suggestion, we conducted expression-based cellular deconvolution analyses using CIBERSort. We found that TL is weakly associated with the proportion of naïve B cell ($r=-0.17$, $p=0.033$), and activated dendritic cells ($r=0.17$, $p=0.033$). However, our findings did not reveal any significant associations between telomere length and immune infiltration after multiple testing corrections ($FDR < 0.1$, Table 1 below). Therefore, based on our current data, we did not find any significant associations between specific immune subsets and telomere restriction fragment (TRF)-based telomere length measurements. Furthermore, WGCNA can indirectly capture immune-related modules if the expression patterns of immune-related genes are correlated within specific co-expression modules. This means that if genes involved in immune functions tend to show similar expression patterns across samples, they may cluster together in a module identified by WGCNA. In our analysis, we did not identify any co-expression module that exhibited a specific association with immune-related modules.

As recommended by other reviewers, we have performed multiple testing corrections for all 110 tested genes, including those associated with telomerase activity, alternative lengthening of telomeres (ALT), and telomere protection (Supplementary Table 8. In the top prioritized list (Table 2 below), we observed moderate significance ($FDR < 0.2$) for *NFX1* and *BM11*, which are linked to telomerase activity, as well as *SMARCAL1*, which is associated with the ALT pathway. However, the results regarding the associations between genes in the ALT pathways and telomere length (TL) were not as robust. Therefore, we have adjusted the emphasis on the findings related to ALT pathways in the revised manuscript. This modification ensures a balanced presentation of the results and reflects a cautious interpretation of the data considering multiple testing corrections. Furthermore, we employed linear regression with the covariates adjusted, and used t-tests to explore the relationships between telomere length and gene expressions. After adjustment for infant sex and maternal age, RTL showed a positive association with *NFX1* expression ($p = 0.013$) in placenta.

Table 1. Correlation Analysis of TL and the proportion of immune cells

cell type	corr	p	fdr
B cells naive	-0.17	0.033	0.256
B cells memory	-0.05	0.484	0.699
Plasma cells	-0.12	0.116	0.382
T cells CD8	-0.10	0.181	0.464
T cells CD4 naive	-0.13	0.085	0.382
T cells CD4 memory resting	-0.04	0.611	0.699
T cells CD4 memory activated	0.11	0.166	0.464
T cells follicular helper	0.04	0.594	0.699
T cells regulatory (Tregs)	-0.05	0.525	0.699
T cells gamma delta	-0.04	0.639	0.699
NK cells resting	-0.07	0.378	0.699
NK cells activated	0.02	0.764	0.799
Monocytes	0.07	0.402	0.699
Macrophages M0	0.05	0.521	0.699
Macrophages M1	0.01	0.866	0.866
Macrophages M2	0.14	0.083	0.382
Dendritic cells resting	0.04	0.572	0.699
Dendritic cells activated	0.17	0.033	0.256
Mast cells resting	0.09	0.232	0.533
Mast cells activated	-0.12	0.111	0.382
Eosinophils	0.05	0.559	0.699
Neutrophils	-0.07	0.404	0.699

Table 2. Correlation Analysis of TL and essential genes (with multiple testing corrected, show top 10 genes here)

Gene	Corr	Category	P -value	P .adjusted	P -value regression	P .adjusted regression
NFX1	0.236	telomerase activity	0.002	0.161	0.006	0.228
SMARCAL1	0.228	ALT	0.003	0.161	0.005	0.228
BMI1	0.220	telomerase activity	0.004	0.161	0.015	0.323
DAXX	0.206	ALT	0.008	0.202	0.005	0.228
TNKS	0.202	telomerase activity	0.009	0.202	0.022	0.346
TOP3A	0.192	ALT	0.013	0.210	0.013	0.323
TPP1	0.188	telomere protection	0.015	0.210	0.021	0.346
RAD50	0.183	telomerase activity	0.019	0.210	0.054	0.346
CGGBP1	0.178	telomere protection	0.021	0.210	0.049	0.346
ATRX	0.177	ALT	0.022	0.210	0.048	0.346

4. The authors state that "Since newborn TL could predict later life TL, it is critical to investigate TL determinants of the newborn. However, no existing study has harnessed a significant number of early samples without postnatal environmental exposure to study TL." It would be helpful to give some additional context. For instance, TL has been measured in newborn bloodspots (Guthrie cards) and in circumcised foreskin. From a biological impact perspective, studies have shown that longer genetically-predicted TL is associated with elevated risks of cancer, including two studies that observed an effect in childhood cancers (PMIDs: 33115534, 31525475). Notably, both showed that the effect was stronger in adolescent-onset cancer than in those arising before age 12, supporting your assertion to the importance of studying early-life TL.

#Response: Thanks for providing relevant studies that support the conclusions drawn in our research. These studies have significantly enriched the discussion section of our manuscript, and we have cited and discussed these findings to reinforce the validity and robustness of our own conclusions. "Recent studies have reported that longer telomere length is associated with an increased risk of adolescent-onset ependymoma and osteosarcoma. These findings underscore the importance of studying telomere length regulation in early life and illuminate future research on the potential consequences of genetic effects that may lead to developmental disorders and diseases by influencing telomere length". By incorporating these additional references, we have been able to present a more comprehensive and well-supported discussion of our research outcomes. The cited studies have complemented our own findings and added valuable context to the broader scientific landscape in which our work is situated.

5. The authors show that "The estimated PRS score of TL was significantly correlated with placental TL measured in our study". While the study is not powered for a GWAS of placental telomere length, it is necessary to include a supplemental table showing the SNP-level associations for each of the ~200 LTL GWAS hits with placental aTL and/or RTL (P, Beta, SE, EA, EAF). Since your sample is Asian, it would be okay to only do this more GWAS hits from the trans-ethnic analysis that showed nominal ($P < 0.05$) association in the SCHS study.

#Response: Thanks for the comment. In our analysis, the associations with PRS and TL are not tested in individual SNP, but used a summarized score instead, which is described in the methods of calculating PRS. As suggested, we included a supplemental table (Supplementary Table 7 showing the SNP-level associations for each of the ~200 LTL GWAS hits with placental RTL (P, Beta, SE, EA, EAF). Additionally, we estimated PRS score of TL based on GWAS hits that showed nominal ($P < 0.05$) association in the SCHS study. The result showed that PRS is still significantly associated with TL ($R = 0.18$, $p = 0.023$, see Figure below).

In our study, we did not individually test the associations between Polygenic Risk Score (PRS) and telomere length (TL) for each SNP. Instead, we utilized a summarized score, as detailed in the methods section for calculating PRS. This approach allowed us to consider the cumulative

effect of multiple SNPs when calculating the PRS, providing a more holistic assessment of the genetic influence on TL. Furthermore, when estimating the PRS score for TL, we based it on GWAS hits that exhibited nominal ($P\text{-value} < 0.05$) associations in the SCHS study. Even with this conservative approach, our analysis demonstrated a significant association between the PRS and TL ($R = 0.18$, $P = 0.023$).

Minor:

7. "Since 91.5% of sentinel variants of TL GWAS loci are located in the non-coding genomic region, investigating their regulatory potential on gene expression would accurately determine TL." The authors should correct this to "would accurately determine telomere regulation". Or, if they are implying that this is to improve their prediction models, then say "would improve accuracy of genetic/transcriptomic-based telomere length prediction".

#Response: Thanks for the comment. We have addressed the issue and made the necessary corrections in the revised manuscript: "Since 91.5% of sentinel variants of TL GWAS loci are located in the non-coding genomic region, investigating their regulatory potential on gene expression would improve the accuracy of genetic/transcriptomic-based telomere length predictions."

8. The *in vitro* experiments in the results section have too much description of the genes, which should be relegated to the discussion.

#Response: Thanks for the comment. We have taken your feedback into consideration, and in response, we have made the necessary adjustments to our manuscript. Specifically, we have relocated the detailed description of the genes observed in the *in vitro* experiments from the results section to the discussion section.

9. In Table S4, it would be good to add the EAF as a column.

#Response: Thanks for the comment. We have added the EAF as a column in Table S3-4.

10. You state "A total of 222 sentinel variants (>10 Mb between sentinels)". Did you mean 10kb and not 10Mb? 10Mb seems too far apart, and I note that some of the variants are closer than 10Mb in your table. Also, the distance is really not as important as the LD (R^2) between two SNPs, and is a better way to prune.

#Response: We sincerely appreciate the comment provided. To optimize our results, we applied a distance of 10kb and set a threshold of $R^2 \leq 0.001$ for the pruning process. As per your suggestion, we have meticulously incorporated the necessary corrections in the revised manuscript to ensure the clarity and validity of our methodology.

11. Fig 1A or Fig 1B should show which layer (fetal, middle, maternal) progressed to downstream TRF and RNA analysis. Also, was the fetal layer from all 4 sampled sections included in TRF analysis for all subjects, and then averaged to yield aTL? Or was one sample used for TRF, another for genotyping, another for RNAseq, etc.?

#Response: We appreciate the reviewer's feedback. In response, we have revised the methodology section to provide more detailed information on the experimental procedures. In order to visualize the placenta sampling method used in this study more clearly, we revised the corresponding flowchart (Fig 1A and 1B). One random sample was selected from the 4 sections in fetal layer samples for subsequent procedures, including TRF assay, genotyping, and RNA-seq.

Reviewer #3 (Remarks to the Author):

In this work Chang et al aim to dissect the causal genes and transcriptomic determinants of TL GWAS loci. This is certainly a gap in our current knowledge around TL genetics – much of the GWAS work is in adult leukocyte TL, and as the authors state TL dynamics are a combination of genes and environment with age attrition. While the premise of the work and the measurement of TL/RNASeq in placental tissue is added value, this work suffers from a lack of methods details that make it challenging to evaluate scientific rigor.

#Response: We extend our heartfelt gratitude to the reviewer for the positive and constructive comments, which will aid us in refining the details of our methodology description and ensuring the scientific validity of our results can be effectively evaluated.

There are some interesting results: honing in on 31 causal genes using three parallel approaches and gene expression in the placental tissue is exciting as it includes some previously undescribed targets); and the follow up validation of four novel genes are important additions to our understanding of the genomic underpinnings of TL. However there are a number of specific concerns as follows that need to be clarified and addressed first:

1. The methods state “To analyze the possible intraplacental variation of TL, all the processed biopsies were divided into three parts and sampled at the fetal and maternal layers to obtain eight samples of each placenta, as described by Wyatt et al. “. However, Wyatt et al describe 9 samples of each placenta, and not 8. All additional methods re TL measurement with TRF and RNASeq quantification and analysis only describe 166 placental samples, i.e. the unit being each single placenta source (166 pregnancies). It is not possible to know how the 8(or 9) samples were handled in the analysis, and which sample the RNASeq data corresponded to. Figure 1 C looks like the 8 TL measurements from one sample, but the data re not described with respect to all 166 placenta with 8 (or 9) samples. Sup Fig 1 shows 15 samples randomly selected, but are they all from the same region across 15? In general this is all poorly described and difficult to evaluate – the methods need to be clear on exactly how many samples from each placenta were included, which sample the RNASeq was generated on, and how all these data were processed in each analysis step in methods. The gel plots are not useful by themselves, the TL data needs to be included in tables for comparison.

#Response: We appreciate the reviewer's feedback. In response, we have revised the methodology section to provide more detailed information on the experimental procedures. In order to visualize the placenta sampling method used in this study more clearly, we revised the corresponding flowchart (previous Fig 1A and 1B).

First of all, we admire your keen insight. As you pointed out, a reference to placental sampling was inadvertently inserted, which we have corrected in this revision. We optimized the placenta sampling protocol recommended in the Amsterdam Placenta Workshop Group Consensus

Statement to allow standardized tissue collection for this study [1]. Specifically, four equidistant sampling points were selected from a hypothetical concentric area with a radius of 2 cm centered at the placental umbilical cord insertion point to collect a 1.5 × 1.5 cm full-thickness placental biopsy. The placenta was placed with the fetal side up and oriented with the largest umbilical artery on the fetal side of the placenta as a reference. To avoid contamination with cells of non-target origin, the membranes were excised and excess blood was removed using sterile filter paper. A processed full-thickness biopsy was divided into three equal parts, and samples were taken from both sides, located in the fetal and maternal layers. A total of eight samples were obtained per placenta. All samples were stored in RNAlater at −80 °C until extraction. To analyze the potential intraplacental variation of TL, one random sample from the four fetal layer-derived biopsies collected from each of the 166 placentas, was selected for subsequent testing including TRF assay, genotyping, and RNA-seq (see Figure below). We have polished sample processing descriptions in our revised manuscript.

In addition, by randomly sampling of ten placental samples, we performed repeated measurements on different regions of the same placental sample for additional three more times (see some examples of Southern blot in the Figure below, marked as A). This approach allowed us to evaluate the homogeneity of telomere status across the entire placenta and provided valuable insights into potential measurement and sampling errors. Our current results showed that the quantitative analysis error did not exceed 10% between different experimental batches (B). This consistency demonstrates the reliability of our TRF measurements. To further ensure the reliability of our data analysis, we utilized the TeloTool software with a strict Fit threshold setting of 60%. This stringent criterion allowed us to include only length determinations meeting the predefined standards in our final results, minimizing any potential bias or errors

during data analysis (C). We have added some descriptions and results for this verification in our revised manuscript.

A. Examples of southern blot was conducted to analysis the telomere length of samples obtained from the placenta; B. The statistical analysis presents the telomere length measurements of the same placental sample obtained from different positions and different batches of measurements; C. Telomere length analysis was performed using TeloTool software with a threshold set at 60%. The black bars represent the analyses that met the fitting criteria, while the red bars represent the analyses that did not meet the fitting criteria and will be excluded from the subsequent statistical analysis.

In addition, we have included a table (Placental_TL_TRF_norm.txt.gz) in figshare that provides complete TRF data for each individual: <https://figshare.com/s/f6de1a56ad7c448c1f4c>.

1. Khong TY, Mooney EE, Ariel I, et al. Sampling and Definitions of Placental Lesions: Amsterdam Placental Workshop Group Consensus Statement. Arch Pathol Lab Med. 2016;140(7):698-713. doi:10.5858/arpa.2015-0225-CC

2. How is TL compared between GTEx and these data when the underlying assay itself was different? The methods do not describe this comparison and the ability to even do so. Is relative TL standardized between the two datasets? With no articulation of the methods and approach, it is not possible to assess any inferences regarding TL in placenta to adult tissue in GTEx.

#Response: Previously, in both our study and the GTEx dataset, we utilized relative telomere length (RTL) calculations that involved comparing observed telomere length to internal controls. In our study, HeLa DNA was introduced as an internal control in each Southern blot gel, and RTL was estimated by determining the ratio of telomere length in the sample to that of HeLa DNA. In GTEx, RTL was expressed as the telomere quantity index (TQI), which is derived by comparing the telomere probe intensity to the intensity observed from the same quantity of reference DNA sample (ALK). Both of these normalization methods were designed to account for plate-to-plate variations. Within the GTEx dataset, RTL measurements obtained

from Luminex RTL showed a strong correlation with telomere length measured by Southern blot ($r = 0.7$ and an r^2 of approximately 0.5). Additionally, a study assessing the relationship between Southern blot and Luminex assays in paired samples reported a linear association between these two measures [1].

However, as the reviewer pointed out, our relative telomere length (RTL) assay and the GTE_x Luminex assay represent different methods of measuring telomere length. Importantly, the internal reference standards on which the calculation of RTL is based also vary. Therefore, direct comparison of RTL between the two studies may not be appropriate. Various studies have reported the average telomere length in different tissues [2]. For instance, early research suggested that bone marrow stem cells and germ cells tend to have relatively longer telomeres, while other tissues such as blood cells and skin cells have shorter telomeres. In this revision, to fairly assess whether there is a significant difference between placental telomere length and that of other human tissues, we collected additional samples from four different human tissues, excluding the placenta, and conducted telomere restriction fragment (TRF) analysis. Our result revealed that placenta (the mean telomere length was 11.83 kb ranging from 7.95 kb to 17.85 kb across all samples (N = 166)) telomere is longer than blood (the mean telomere length was 9.44 kb ranging from 4.52 kb to 14.66 kb across all samples (N = 231)), skin (8.73 ± 4.39 kb, N=12, (Mean \pm SEM)), heart (8.69 ± 3.18 kb, N=6) and lung (9.15 ± 4.97 kb, N=6).

Left: The telomere length of different tissues, lane 1, molecular markers, lane 2-3, blood samples; lane 4-6, skin samples; lane 7-9, heart samples and lane 10-12, lung tissues.

Right: Mean telomere lengths were quantified in placenta and other 4 human tissues. For boxplots, 5-95 percentile and outliers are shown.

1. Pierce, B. L., Jasmine, F., Roy, S., Zhang, C., Aviv, A., Hunt, S. C., Ahsan, H., & Kibriya, M. G. (2016). Telomere length measurement by a novel Luminex-based assay: a blinded comparison to Southern blot. *International journal of molecular epidemiology and genetics*, 7(1), 18–23.

2. Kathryn Demanelis, Determinants of telomere length across human tissues, science, 2020, 369(6509)

3. How do the effect sizes in the three GWAS used for the meta-analysis compare and is the fixed effects meta-analysis suitable. There is correlation in the measurements (telseq from TOPMed), qPCR and Southern blot, but there is no discussion on how this affects the meta-analysis and the choice used for the fixed effects. Additionally, TOPMed was in itself a meta-analysis, with released ancestry stratified results publicly released. It is not clear what was used here.

#Response: We admitted that the accuracy of meta-analysis can be attenuated in the presence of cross-study heterogeneity. In this study, we evaluated heterogeneity using the I^2 statistic, it can reflect the degree of variation in effect sizes between studies. The assessment of heterogeneity between cohorts reveals that it is nonsignificant for the majority of variants (Supplementary Table 3). To further strengthen our analysis, we have conducted a random-effect meta-analysis for the lead SNPs, enabling us to compare the results with the fixed-effect approach. The findings from the random-effect meta-analysis demonstrate reassuringly concordant effect sizes (see Table below) for most variants, even including those with higher heterogeneity and discrepant P-values. This alignment of effect sizes reinforces the robustness and consistency of our results across different cohorts, providing additional confidence in the validity of our findings.

In the discussion section, we discussed the issue of heterogeneity: “The accuracy of meta-analysis can be attenuated in the presence of cross-study heterogeneity, which can be attributed to several factors. One significant consideration is the meta-analysis of multiple TL studies, which might utilize slightly different definitions of the phenotype under investigation. Consequently, the effect sizes across these studies may vary. Additionally, the presence of ancestry-specific effects can further contribute to the observed heterogeneity. Together, these factors highlight the importance of carefully interpreting GWAS results in the context of diverse study populations and varying definitions of phenotypes, thereby emphasizing the need for rigorous validation and comprehensive understanding of the underlying genetic associations of TL.”

In response to your last concern, specifically, we used the meta-analysis results of TOPMed in this study.

Chr:Pos	dbSNP ID	Nearest genes	EA/NEA	Fixed model			Random model		
				Beta	SE	P-meta	Beta	SE	P-meta
1:11778977	rs55967531	C1orf167	A/G	-0.014	0.0024	8.81E-09	-0.014	0.003	5.40E-06
1:23274086	rs678007	LINC01355	T/C	0.0103	0.0018	1.80E-08	0.011	0.003	0.0001326
1:185275436	rs10798002	SWT1	A/C	0.0106	0.0015	1.34E-12	0.011	0.002	1.34E-12
1:212802436	rs4951639	TATDN3	T/C	0.0086	0.0016	3.40E-08	0.009	0.002	3.40E-08

RP11-									
1:226424585	rs12032713	118H4.3	T/C	0.0168	0.0026	1.81E-10	0.023	0.008	0.002738
2:16449516	rs4832615	AC010880.1	T/C	0.01	0.0017	9.95E-09	0.010	0.002	7.00E-06
2:65330153	rs871974	SPRED2	T/C	-0.0095	0.0017	1.18E-08	-0.010	0.002	1.18E-08
3:133684452	rs77247608	TFP1	A/G	0.0094	0.0016	7.07E-09	0.009	0.002	7.07E-09
6:29696535	rs3129087	ZFP57	T/C	0.0123	0.002	4.17E-10	0.012	0.002	4.17E-10
7:50295636	rs4917017	SPATA48	A/G	0.0096	0.0016	1.63E-09	0.010	0.002	1.63E-09
10:32337541	rs2505400	EPC1	A/G	-0.0116	0.002	1.52E-08	-0.012	0.002	1.52E-08
10:133232519	rs3008359	VENTX	T/G	0.0132	0.0021	4.87E-10	0.013	0.002	4.87E-10
11:118720034	rs503542	AP002954.1	A/G	-0.0098	0.0015	3.50E-11	-0.010	0.002	3.47E-06
13:44479381	rs9533871	TSC22D1	T/C	0.0081	0.0015	3.58E-08	0.008	0.001	3.58E-08
14:72805970	rs2332919	DPF3	T/C	-0.0105	0.0016	1.46E-10	-0.010	0.002	2.64E-09
14:96481626	rs138980163	AK7	A/G	0.013	0.0022	8.11E-09	0.013	0.002	8.11E-09
14:103936818	rs10220464	TDRD9	A/G	0.0086	0.0015	2.45E-08	0.011	0.004	0.00333
17:39552588	rs7503377	CDK12	T/C	0.0109	0.0019	1.50E-08	0.011	0.002	3.06E-06
18:76984378	rs470550	MBP	T/C	-0.0095	0.0015	1.32E-10	-0.010	0.001	1.32E-10
22:31617379	rs7293143	SFI1	A/G	-0.0179	0.003	2.44E-09	-0.018	0.003	2.44E-09

4. The PRS approach is described as “summing over all SNPs meeting a set of thresholds, respectively”. What were these thresholds, all SNPs meeting a threshold cannot be included without at a minimum pruning on LD to minimize overfitting. Again – without details, it is not possible to evaluate scientific rigor. Also this statement contradicts line 164 in results which states “polygenic risk score (PRS) analyses based on the sentinel significant variants of all independent loci from TL meta-analysis results”. PRS performance should not be tested in the dataset from which it was derived – i.e. UKBB.

#Response: We thank the reviewer’s suggestion. We now added more details about PRS approach, and made corrections for the statement in the revised manuscript.

1) “Polygenic scores of TL were constructed using PRSice-2 to gauge the associations between reported variations of TL in general populations and in the current study. The scores were computed as the weighted sum of effect allele dosages, as a matrix multiplication of SNP dosages per individual by betas per SNP, i.e., the outcome is a single score of each individual’s genetic loading for TL. Our measure of predictive power is the incremental R^2 from adding the score to a regression of the phenotype while adjusting for top five genotyping PCs, sex, and maternal age. The PRS was calculated by summing over all SNPs meeting a set of thresholds, respectively. We used the default P -value thresholds in PRSice-2 (from $5e-8$ to 0.5 , step size: $5e-5$). All SNPs that met the specified threshold underwent LD pruning to reduce overfitting, utilizing the default settings (distance for clumping: 250kb, r^2 threshold: 0.1). The null P -value of the association of the best-fit GWAS P -value threshold was converted to the empirical P -value under 10,000 permutations. Pearson’s correlations between PRS and RTL were used to compare the PRS analytical performance for the Chinese samples in UKBB, all samples in GTE_x, and all samples in this study”.

2) We have changed “polygenic risk score (PRS) analyses based on the sentinel significant

variants of all independent loci from TL meta-analysis results” to “polygenic risk score (PRS) analyses based on TL meta-analysis results”.

3) We have removed the results of testing PRS performance in the UKBB from both Figure 2 and the main text of the manuscript.

5. Coloc was run, this assumes a single GWAS/eQTL causal signal. This is known to not be true for many of the GWAS loci for TL. Other approaches that do not make this assumption are more appropriate. Additionally, $P_4 > 0.5$ seems liberal – please justify. There should generally be a higher PPH4, and also a criteria on PPH3 that is applied for coloc.

#Response: Thanks for the comment, we assumed that the GWAS/eQTL sharing a single causal signal in our study. It is important to consider that when accurate study-level linkage disequilibrium (LD) information is not available, the misrepresentation of LD between causal variants can significantly impact the accurate identification of multiple causal variants in a given region. Therefore, in such situations, relying on the assessment of a single causal variant becomes the most reliable approach. This approach ensures a more robust and accurate determination in the absence of precise study-level LD information (Foley, C. N, et.al, 2021). To enhance the reliability of our results, we have adjusted the coloc threshold. We provide results of colocalization with strong evidence and likely colocalization with suggestive evidence using a rigid standard ($PP_4 \geq 0.8$ and $PP_4/PP_3 \geq 5$) and a liberal standard ($PP_4 \geq 0.5$ and $PP_4/PP_3 \geq 3$). By employing these criteria, we aim to maintain rigorous standards for identifying coloc genes while also considering potential associations that may exhibit lower confidence but warrant further investigation. Given that we consider the intersection of COLOC and TWAS/SMR results, we utilize suggestive evidence from COLOC to enhance the discovery rate of potential causal genes. This consideration balances the rigor of our analysis with the aim to uncover novel genetic contributors to the phenotype of interest. Finally, we defined likely causal genes related to TL as the intersection of gene prioritization results from COLOC (suggestive evidence) and the union of TWAS and SMR, finally we retrieved 23 likely causal genes related to TL (Supplementary Table 12).

1. Foley, C. N., Staley, J. R., Breen, P. G., Sun, B. B., Kirk, P. D. W., Burgess, S., & Howson, J. M. M. (2021). A fast and efficient colocalization algorithm for identifying shared genetic risk factors across multiple traits. *Nature communications*, 12(1), 764. <https://doi.org/10.1038/s41467-020-20885-8>

6. Line 150: How was heterogeneity defined? And, does this statement mean that 13% of the loci that were not identified in the meta-analysis because of heterogeneity between the studies? How much of this could be due to different measurement assays in the GWAS themselves? See point #3 above?

#Response: Thanks for pointing out this issue again. We admitted that the accuracy of meta-analysis can be attenuated in the presence of cross-study heterogeneity. Our meta-analysis successfully validated 78% of the loci initially reported in the three previous GWAS studies. However, due to potential observed heterogeneity, the remaining loci did not reach statistical significance. In this revision, we evaluated heterogeneity using the I² statistic, it can reflect the degree of variation in effect sizes between studies. The assessment of heterogeneity between cohorts reveals that it is nonsignificant for the majority of variants. To further strengthen our analysis, we have conducted a random-effect meta-analysis for the novel SNPs, enabling us to compare the results with the fixed-effect approach. The findings from the random-effect meta-analysis demonstrate reassuringly concordant effect sizes for most variants, even including those with higher heterogeneity and discrepant P-values. This alignment of effect sizes reinforces the robustness and consistency of our results across different cohorts, providing additional confidence in the validity of our findings.

7. The idea of building upon the PRS with additional data including transcriptomics is interesting. Can this be validated in additional datasets?

#Response: We are grateful for the insightful comment provided by the reviewer. In our previous manuscript, we validated and explored the potential of enhancing polygenic risk scores (PRS) with additional data, specifically transcriptomics information from the GTEx dataset. However, we were unable to locate a more suitable additional dataset that contained all the required data elements (genotype, gene expression, and telomere length). As suggested by the reviewer, to validate our telomere length prediction strategies in an additional cohort, we chose to utilize the UK Biobank Chinese data, which provides both genotype and telomere length information.

To predict gene expression, we employed a strategy akin to the one used in Transcriptome-Wide Association Studies (TWAS). Specifically, we leveraged the weights obtained from telomere length expression quantitative trait loci (TL eQTLs) to predict gene expression from the available genotype data in the UKBB Chinese dataset. In this model, we captured several important predictors such as STN1, SMIM12, and TSSC1. The overall correlation between the predicted telomere length (TL) and the actual TL values was found to be significant (see Figure below, $R = 0.35$, $p < 2.2e-16$). These results highlight the effectiveness and reliability of incorporating both PRS and transcriptomics information in predicting TL.

Performance of TL prediction model in UKBB. Scatter plot shows the actual TL values in UKBB against the values predicted by the model.

REVIEWERS' COMMENTS

Reviewer #2 (Remarks to the Author):

The authors have gone to great lengths to address reviewer comments and to adjust the presentation of results to be better aligned with the inferences they are (and are not) able to reach with these new data. I commend them for their thorough response and have no further concerns.

Reviewer #3 (Remarks to the Author):

The authors have been highly responsive to all major concerns raised in my prior critique. I believe the additional analysis on TL assays, and clarification of analysis (either dropping results or reanalysis where needed with appropriate changes to pipeline) are suitable. I have one pending concern.

There is discrepancy in reference to the heart and lung tissue used for TL comparisons. Methods describe these as fetal samples, but results describe these as adult tissues. The conclusion statements around the observed patten in Fig 1 that heart and lung are as low (or lower) than blood and skin are therefore not interpretable. This needs to be addressed.

Dear Editor,

We thank you and reviewers for evaluating our manuscript again (NCOMMS-23-02741A). Here, we have carefully addressed your and the reviewers 3' additional comments below and in the manuscript accordingly. Accompanying this letter, please find the revised version of our manuscript.

REVIEWER COMMENTS

Reviewer #2 (Remarks to the Author):

The authors have gone to great lengths to address reviewer comments and to adjust the presentation of results to be better aligned with the inferences they are (and are not) able to reach with these new data. I commend them for their thorough response and have no further concerns.

#Response: We sincerely appreciate the reviewer for the positive remarks.

Reviewer #3 (Remarks to the Author):

The authors have been highly responsive to all major concerns raised in my prior critique. I believe the additional analysis on TL assays, and clarification of analysis (either dropping results or reanalysis where needed with appropriate changes to pipeline) are suitable. I have one pending concern.

There is discrepancy in reference to the heart and lung tissue used for TL comparisons. Methods describe these as fetal samples, but results describe these as adult tissues. The conclusion statements around the observed pattern in Fig 1 that heart and lung are as low (or lower) than blood and skin are therefore not interpretable. This needs to be addressed.

#Response: We are deeply appreciative of the reviewer for agreeing with our previous revisions. With respect to the raised concern, we acknowledge an inaccuracy and typos in our description of the origin of heart, liver, skin, and blood samples. It is important to clarify that these tissue samples were sourced from adults, not embryos. We recognize that a significant factor contributing to this discrepancy was a misunderstanding in communication among the different research groups. We sincerely thank you for bringing this to our attention, and we have accordingly revised the relevant sections in the revised manuscript to ensure the accuracy of our methodology.

As we described in the results section, previous studies have reported that the Terminal Restriction Fragment (TRF) length of the telomere is highly synchronized across various tissues at birth. In relation to our conclusions, we observed that the average telomere length in placental

tissue is longer than in other adult tissues. This finding is consistent with our expectations, indicating that telomere length is associated with cellular senescence, peaking at birth and declining with age and exposure. This result reaffirms the connection between telomere length and cellular aging, underscoring its significance in understanding the cellular aging processes associated with age and environmental exposure.

We have updated the relevant descriptions in the related sections (abstract, results, and methods) and highlighted the source of our samples in the conclusion to ensure accuracy throughout the manuscript. We thank you for your patience and guidance, and we remain committed to guaranteeing the consistency and reliability of our study in both methods and conclusions.